# Neural Message-Passing on Attention Graphs for Hallucination Detection

**Fabrizio Frasca**[*]
Technion
`fabriziof@campus.technion.ac.il`

**Guy Bar-Shalom**[*]
Technion
`guybs99@gmail.com`

**Yftah Ziser**
University of Groningen, NVIDIA Research

**Haggai Maron**
Technion, NVIDIA Research

## Abstract

Large Language Models (LLMs) often generate incorrect or unsupported content, known as hallucinations. Existing detection methods rely on heuristics or simple models over isolated computational traces such as activations, or attention maps. We unify these signals by representing them as *attributed graphs*, where tokens are nodes, edges follow attentional flows, and both carry features from attention scores and activations. Our approach, CHARM, casts hallucination detection as a graph learning task and tackles it by applying GNNs over the above attributed graphs. We show that CHARM provably subsumes prior attention-based heuristics and, experimentally, it consistently outperforms other leading approaches across diverse benchmarks. Our results shed light on the relevant role played by the graph structure and on the benefits of combining computational traces, whilst showing CHARM exhibits promising zero-shot performance on cross-dataset transfer.

## 1 Introduction

Despite their impressive capabilities, LLMs frequently produce outputs that are factually inaccurate, logically inconsistent, or unsupported by the input context, broadly referred to as *hallucinations* (Pagnoni et al., 2021; Cao et al., 2022; Qiu et al., 2023). As LLMs are increasingly applied in diverse domains, detecting hallucinations becomes crucial for ensuring their safe and reliable use. This phenomenon is inherently complex and multi-faceted, and methods for *automated hallucination detection* (HD) have recently received significant attention (Yin et al., 2024; Bar-Shalom et al., 2026).

A straightforward approach for HD is to query LLMs multiple times, either by asking them to judge their own outputs (Kadavath et al., 2022) or by sampling alternative generations to measure semantic variability (Kuhn et al., 2023). While effective in some cases, this strategy requires repeated rollouts, making it both slow and computationally expensive, and thus unsuitable for real-time or large-scale use. A more scalable line of work leverages the internal signals produced by LLMs during decoding, which we refer to as *computational traces*. In particular, most works focus on *linearly* probing residual stream activations on selected layers and token positions (Orgad et al., 2025; Azaria & Mitchell, 2023; Belinkov, 2022). More recently, attention maps have shown to provide an additional perspective on model behaviour, e.g., by leveraging prompt-response attention ratios (Chuang et al., 2024). Although providing meaningful, alternative cues on hallucinations existing attention-based techniques rely on simple models or handcrafted *heuristics* (Sriramanan et al., 2024; Binkowski et al., 2025). Furthermore, all the above methods treat computational traces in isolation, despite capturing complementary aspects of hallucinations. To date, a systematic exploration of the interplay of computational traces is still lacking. More broadly, the field currently lacks a framework applying modern deep learning techniques to structured, holistic representations of computational traces, leaving the community to rely on heuristic, single-signal approaches.

In this paper, we propose a unified framework that represents LLM computational traces as *attributed graphs*, a natural, yet under-explored perspective in the HD literature. Similarly to recent works

---

[*]Equal contribution.

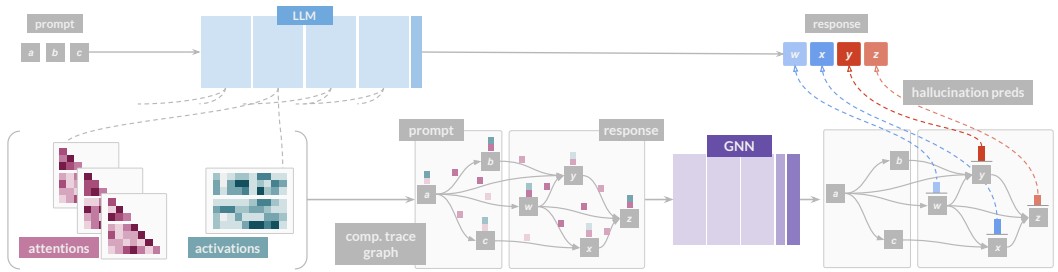

Figure 1: **Overview of CHARM.** We extract attention and activation matrices from LLM computations and build an attributed graph from them: edges and their features are derived from off-diagonal attention scores; node features are based on activations, and diagonal attention values. The resulting graph is processed by a GNN-based architecture, which outputs either token-level hallucination scores (as illustrated) or a global hallucination score for the entire sentence.

dealing with the analysis of learnt attention computational flows (Barbero et al., 2024; El et al., 2025), this formulation considers tokens as nodes and draws connections between them based on the structure of attention maps calculated during text generation. Crucially, both nodes and edges can be endowed with features derived from the values of computational traces across layers (and heads): node features capture token-wise signals such as activations and self-scores (the attention a token assigns to itself), while edge features encode pairwise interactions, most prominently the attention between distinct tokens. This perspective casts HD as a *graph learning* problem, which has recently obtained successes in broad-ranging domains (Monti et al., 2019; Gonzalez et al., 2021; Liu et al., 2023) and, we argue, is well suited to this task. First, representing computational traces as attributed graphs allows to naturally integrate heterogeneous signals, which may hold varying predictive value across generation tasks. Second, the framework accommodates different levels of detection granularity, with the standard setup of graph classification corresponding to response-level detection, and that of node classification to the token-level one. Finally, this formulation directly leverages the rich body of work on Graph Neural Networks (GNNs) and their code-libraries (Fey & Lenssen, 2019), providing principled and well-studied tools for tailored HD models.

Motivated by these advantages, we introduce CHARM (see Figure 1), an HD approach based on a Graph Neural Network operating on computational trace graphs (Gilmer et al., 2017). Our framework can jointly process different computational traces, and subsumes known detection heuristics: we prove that it can express recent attention-based methods (Chuang et al., 2024; Sriramanan et al., 2024) either at the token or response granularity levels. Experimentally, CHARM *consistently outperforms* these heuristics, as well as other leading methods across benchmarks and detection resolutions. Our analyses further reveal that incorporating activations into computational trace graphs alongside attention-features may improve detection of non-contextual hallucinations. Beyond state-of-the-art comparisons, our ablation studies demonstrate the importance of the graph structure, the main principle driving CHARM. Finally, we report promising zero-shot cross-dataset transfer results and observe robustness to graph sparsifications, indicating viable trade-offs between accuracy and efficiency.

**Contributions** are summarised as follows. (1) We introduce a unified view of LLM *computational traces* as *attributed graphs*, where tokens are nodes connected by attention-induced edges, and both nodes and edges are enriched with features such as activations and attention scores. (2) We introduce CHARM, which casts HD as graph learning on computational trace graphs. It uses a GNN that provably subsumes attention-based heuristics, opening new application frontiers for machine learning on graphs. (3) We show that CHARM consistently outperforms leading HD methods across diverse benchmarks and granularities, while exhibiting promising zero-shot transfer capabilities.

## 2    RELATED WORK

**Hallucinations and their detection in LLMs.**    The term "hallucinations" in LLMs broadly refers to errors in text generation where outputs are unfaithful to the input or external facts (Orgad et al., 2025). These include knowledge inaccuracies, flawed reasoning, biases, and references to non-existing (Liu et al., 2022; Huang et al., 2023; Ji et al., 2023; Rawte et al., 2023) . Hallucinations can involve

complex failures and manifest in subtle ways, including at the granularity of single tokens (Orgad et al., 2025). Early detection approaches leverage uncertainty measures in next-token prediction or semantic consistency of responses (Kadavath et al., 2022; Varshney et al., 2023; Kuhn et al., 2023; Manakul et al., 2023). Alternatively, recent work propose detectors on LLM computational traces; prominent examples include (hidden) activations (Kadavath et al., 2022; Snyder et al., 2024; Yuksekgonul et al., 2024; Zou et al., 2023; Yin et al., 2024; Chen et al., 2024; Simhi et al., 2024; Li et al., 2024; Marks & Tegmark, 2024; Burns et al., 2023; Rateike et al., 2023; Bar-Shalom et al., 2025) and attention matrices (Sriramanan et al., 2024; Chuang et al., 2024; Binkowski et al., 2025; Bazarova et al., 2025; Zhang et al., 2023). Different traces may be more or less informative for different types of hallucinations: e.g., attention-based heuristics have been evidenced to be predictive in contextual hallucination settings (Chuang et al., 2024). Existing methods mostly rely on heuristics or simple classifiers on specific traces; some are also constrained to coarse detection levels (e.g., whole responses) (Sriramanan et al., 2024; Binkowski et al., 2025; Kuhn et al., 2023).

**Attention-based HD.** Irregular or skewed attention behaviours have been observed to often signal pathological text generation (Xu et al., 2023; Chuang et al., 2024; Binkowski et al., 2025; Bazarova et al., 2025). E.g., hallucinated translations may exhibit localised scores on narrow context windows (Xu et al., 2023); hallucinations in contextual question answering may correlate with excessive focus on response tokens w.r.t. context ones (Chuang et al., 2024). The recent *Lookback Lens* proposes a detection feature based on this intuition. Other works extract spectral or structural features from attention matrices, e.g., via graph Laplacians combined with logistic regression (Binkowski et al., 2025). These approaches, however, remain limited by fixed heuristics and shallow classifiers. Our work generalises this line by employing attention matrices to construct attributed graphs, a formulation that supports predictions at multiple levels of granularity, integrates additional computational signals and unlocks the application of modern (graph-based) deep learning techniques.

**Graphs of LLM computation and Graph Neural Networks.** Recent works have applied graph-theoretic perspectives to neural computations (Vitvitskyi et al., 2025), often graphs induced by attention matrices (Barbero et al., 2024; El et al., 2025). Notably, Barbero et al. (2024) analyse signal propagation on attention graphs, uncovering phenomena such as representational collapse and *oversquashing* (Alon & Yahav, 2021; Topping et al., 2022). These studies highlight the value of attention graphs, but are limited to descriptive and structural analyses. In contrast, we extend attention graphs to more general *attributed graphs* to integrate other computational traces and, importantly, we propose to directly *learn* on these graphs for the task of HD. To this end, we leverage Graph Neural Networks (Kipf & Welling, 2017; Gilmer et al., 2017; Battaglia et al., 2018), a family of architectures which have recently achieved remarkable results in relevant structured domains (Qasim et al., 2019; Monti et al., 2019; Stokes et al., 2020; Gonzalez et al., 2021; Liu et al., 2023).

## 3 LLM COMPUTATIONAL TRACES AS ATTRIBUTED GRAPHS

**Preliminaries.** Throughout this paper, we focus on attention-based, decoder-only LLMs. Abstracting away architectural specifics, we treat them as sharing a common backbone: a stack of transformer-decoder blocks. Let $\mathcal{L}$ denote a reference LLM consisting of $L$ decoder-block layers of $H$ heads each[1], $\vec{p}$ refer to a prompt in input, and $\vec{r}$ to the response $\mathcal{L}$ generates. We consider $\vec{p}, \vec{r}$ to be sequences of tokens of size, resp., $n_p, n_r$ ($n := n_p + n_r$). We use $T_i$ to refer to token at position $i$ in the concatenation $\vec{p} \mid \vec{r}$. Within each transformer block, multi-head attention produces scores, which we collect in attention matrices $A^{l,h} \in [0,1]^{n \times n}$ for layer $l$ and head $h$. Due to the causal structure of decoder-only transformers, these matrices are lower-triangular. For convenience, we define $\boldsymbol{\alpha}_{i,j} \in [0,1]^{L \cdot H}$ as the vector of attention scores between $T_i$ and $T_j$ across all layers and heads. In addition to multi-head attention values, residual stream activations constitute another key source of information about the computation performed by $\mathcal{L}$. For each token $T_i$, we denote by $\mathbf{a}_i^l \in \mathbb{R}^d$ its $d$-dimensional activation vector at layer $l$; this captures the computational state of the model at such token position and processing stage. Together, attention values and activations form the primary signals we use to describe the computational traces of LLMs. While our framework focuses on these two, it can also naturally accommodate additional sources of information, such as logits.

---

[1] One can also consider, without loss of generality, a different number of heads for each layer.

**From computational traces to attributed graphs.** The attention values calculated along the way are, in fact, *pairwise scores* that induce a (non-symmetric) binary relation between tokens. In fact, they define a *directed graph* $G = (V, E)$ on any sequence of tokens $\vec{s} = \vec{p} \mid \vec{r}$, where:

- $V$, the node (vertex) set, is the set of all tokens in $\vec{s}$, namely $\{T_i\}_{i=0}^{n-1}$;
- $E$, the edge set, is the set of ordered pairs $(T_i, T_j)$, $i > j$, signifying $T_i$ attends to $T_j$ in the generation of next tokens: $\alpha_{i,j}^{l,h} > 0$ for some of $\mathcal{L}$'s layers and corresponding heads.

We consider these graphs as *attributed*, in the sense that nodes and edges can host *features* representing $\mathcal{L}$'s computational traces. Edge features are given by the set of attention scores between *distinct* tokens, i.e., $x_{E,(i,j)} = \boldsymbol{\alpha}_{i,j}$'s, with $i \neq j$. Node features are given by the attention scores "paid" by a token to itself, i.e, $x_{V,i} = \boldsymbol{\alpha}_{i,i}$. Node features can also host other token-wise computational traces; we consider residual stream activations $\mathbf{a}_i^l$ at any layer $l$, so that node / token $T_i$ is endowed with feature vector $x_{V,i} = (\boldsymbol{\alpha}_{i,i} \mid \mathbf{a}_i^l)$. Formally, we gather node and edge features in matrices $X_V \in \mathbb{R}^{n \times (L \cdot H + d)}$ (or $\mathbb{R}^{n \times (L \cdot H)}$ should activations be neglected), and $X_E \in \mathbb{R}^{n_E \times L \cdot H}$. The resulting graph is $G = (V, E, X_V, X_E)$. This representation captures both token interactions and per-token computational states, and can be extended to incorporate other traces, e.g., activations from multiple layers or output logits. We leave investigating these aspects to future research endeavours.

**Sparsifying computational trace graphs.** Very small attention scores convey noisy and weak contribution to updating the representation of a token. To reduce computational overhead, we threshold attention scores at $\tau$, zeroing values below it and dropping edges unsupported by any head or layer after this process. In formulae, new graph is defined as $G = (V, E, X_V, X_E^\tau)$, with:

$$(X_E^\tau)_{(i,j),(l,h)} = \begin{cases} 0 & \text{if } \alpha_{i,j}^{l,h} \leq \tau, \\ \alpha_{i,j}^{l,h} & \text{otherwise.} \end{cases} \quad E = \{(T_i, T_j) \mid i > j \text{ and } \exists d \text{ s.t. } (X_E^\tau)_{(i,j),d} > 0\}. \quad (1)$$

As we experimentally show in Section 5.3, sparsifying the graph in this way may significantly improve the efficiency of our yet-to-be-described model, while retaining information about the most relevant token interactions. In the next section we illustrate how, starting from the above formalism, we can instantiate problems such as automated HD as graph learning tasks.

## 4 Neural Message Passing for Hallucination Detection

### 4.1 Hallucination Detection is a Graph Machine Learning Task

**Problem formulation.** Computational trace graphs can be naturally associated with *labels* one seeks to predict for the underlying text generation process. In our specific use-case of HD, these can indeed reflect hallucination annotations at the level of response tokens or the overall response. Concretely, our reference graph $G$ — encoding the computation of $\mathcal{L}$ on prompt $\vec{p}$ and response $\vec{r}$ as per the above Section 3 — can be *annotated* as $(G, y)$, where:

(i) $y \in \{0, 1\}, y = \begin{cases} 1, & \text{if } \vec{r}, \text{ contains } \textit{hallucinating} \text{ passages} \\ 0, & \text{otherwise} \end{cases}$, or

(ii) $y \in \{0, 1\}^{n_r}, y_i = \begin{cases} 1, & \text{if token } T_i, i > n_p \text{ is part of a } \textit{hallucinating} \text{ passage within } \vec{r}, \\ 0, & \text{otherwise} \end{cases}$.

Here, (i) stands for a *graph-wise* label, while (ii) represents labels at the granularity of single (response) tokens. With these premises, we formalise HD as learning a parametric function $f(G) = \hat{y}$ mapping a computational trace graph $G$ to predictions $\hat{y}$ of the corresponding labels $y$. Depending on the task, $\hat{y} \in [0, 1]^k$ with $k = 1$ for graph-wise or $k = n_r$ for token-wise detection.

**Our `CHARM` architecture.** We parameterise $f$ in the family of message-passing Graph Neural Network (GNN) (Kipf & Welling, 2017; Gilmer et al., 2017). These networks are, to date, the de-facto standard for learning on attributed graphs, while possessing an architectural pattern which, as we show next, well aligns them to generalise known approaches. Message-passing networks, in particular, implement local computations reflecting the structure of the input graph, hence benefitting

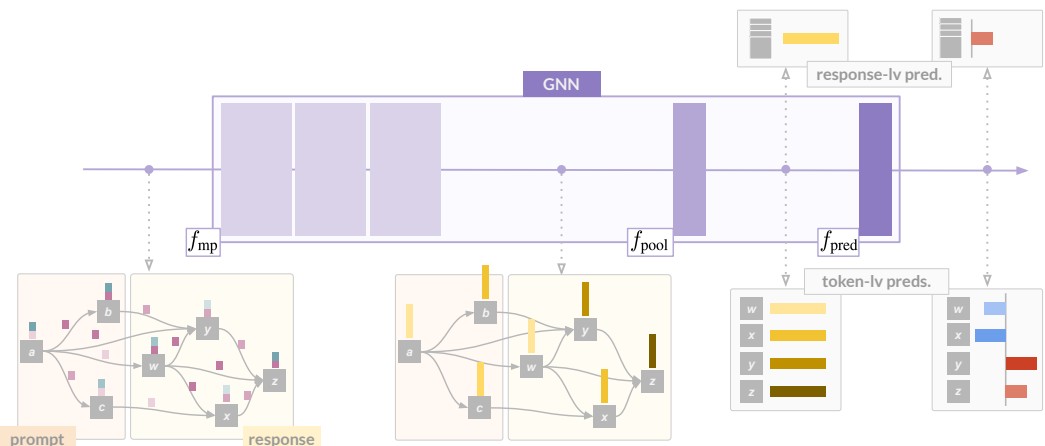

Figure 2: **HD with CHARM.** The input is an attributed graph (shown in the bottom left). First $f_{\text{mp}}$ obtains refined node / token representations via msg-passing. Next, $f_{\text{pool}}$ aggregates these if response level predictions are required. Finally, a projection head, $f_{\text{pred}}$, outputs the detection score.

from the aforementioned sparsification and offering a compelling advantage in terms of computational complexity. In particular, we structure $f$ as: $f = f_{\text{pred}} \circ f_{\text{pool}} \circ f_{\text{mp}}$ (see Figure 2), where:

- $f_{\text{mp}}$ stacks learnable message-passing layers to compute updated token representations from the input graph $G$;
- $f_{\text{pool}}$ aggregates token representations into a single graph representation (e.g., via averaging or summation), or acts as the identity for token-wise detection;
- $f_{\text{pred}}$ applies dense layers to compute graph- or token-wise hallucination predictions.

Starting from the original node (token) features $X_V$, each layer in $f_{\text{mp}}$ calculates and updates hidden node representations by aggregating them (possibly, non-linearly) along the connectivity defined by the attention scores as per Equation (1). That is, the $t$-th layer updates token $i$'s embedding as:

$$h_i^{(t+1)} = \text{up}_t\Big(h_i^{(t)}, \underset{j:\ (i,j)\in E}{\square}\ \text{msg}_t\big(h_i^{(t)}, h_j^{(t)}, x_{E,(i,j)}^\tau, p_{i,j}\big)\Big) \tag{2}$$

where: $\square$ is a permutation invariant[2] aggregator such as sum, average, or max; $x_{E,(i,j)}^\tau$ corresponds to the features of edge $(i,j)$ in $X_E^\tau$; $h_i^{(0)} = x_{V,i}$ are the initial node features; $p_{i,j}$ is a one-hot vector indicating whether edge $(i,j)$ connects prompt to response or response to response tokens. Functions $\text{up}_t, \text{msg}_t$ are parameterised as Multi-Layer Perceptrons (MLPs) running on the concatenation of their arguments. We refer to our overall approach as CHARM, a mnemonic formula for "Catching HAllucinated Responses via (learnable) Message-passing".

This formulation offers two main advantages. First, it provides a unified framework capable of handling both token-level and response-level HD, in contrast to prior approaches that target a single granularity. Second, by leveraging message-passing over computational trace graphs, CHARM can flexibly integrate multiple signals and naturally subsume heuristic-based detectors. As we exemplify next, rather than discarding prior meaningful intuitions, our approach can generalise them.

### 4.2 Expressiveness

Here, we demonstrate how hand-crafted heuristics emerge as special cases of our CHARM. To illustrate this, we focus on two representative methods: (i) Lookback Lens (Chuang et al., 2024), which produces token-wise hallucination scores, and (ii) LLM-Check (Sriramanan et al., 2024), which outputs global sentence-level (graphwise) scores. We show that both heuristics can be approximated, to arbitrary precision and under mild assumptions, by CHARM, highlighting its expressiveness.

---

[2]Permutation invariance ensures that calculated message does not depend on the ordering of nodes in the neighbourhood.

**Lookback Lens** extracts, for each *response* token $i = n_p, \ldots, n_r + n_p - 1$, layer $l$ and head $h$, a heuristic feature $\ell_i^{l,h}$ quantifying the average proportion of attention paid to prompt w.r.t. previously generated response tokens (Chuang et al., 2024). Precisely, `Lookback Lens` outputs scores:

$$P_i^{l,h} = \frac{1}{n_p} \sum_{j=0}^{n_p-1} \alpha_{i,j}^{l,h} , \quad R_i^{l,h} = \frac{1}{i - n_p} \sum_{j=n_p}^{i-1} \alpha_{i,j}^{l,h} ; \qquad \ell_i^{l,h} = \frac{P_i^{l,h}}{P_i^{l,h} + R_i^{l,h}} . \tag{3}$$

where $P_i^{l,h}, R_i^{l,h}$ correspond to the average attention paid by response tokens to, resp., the prompt and the (previously) generated response. We argue that this heuristic can, in fact, be interpreted in the form of message-passing on the (non-thresholded) attention graph described in Section 3, and can thus be captured by our approach:

**Proposition (informal) 1.** *Equipped with a single-layer message-passing stack $f_{mp}$, CHARM running on a non-tresholded computational trace graph ($\tau = 0$) can arbitrarily well approximate token-wise `Lookback Lens` features $\ell_i$ for bounded prompt and response lengths.*

*Proof idea.* The ability to perform such approximation relies on the following: (1) aggregation over previous tokens, as required by $P_i^{l,h}$'s and $R_i^{l,h}$'s, is naturally captured by propagating (and aggregating) attention features on the neighbourhoods of the directed attention graph as per Equation (2); (2) conditioned on mark input $p_{i,j}$, MLP msg can differently route attention features from prompt vs. response tokens in *separate* subspaces of the internal representations; (3) message summation can separately accumulate attention to prompt versus response tokens (*non-normalised* $P_i^{l,h}, R_i^{l,h}$); (4) MLP up can normalise and combine these aggregated scores as required, calculating the ratio in Equation (3). A formal statement of informal proposition 1 is, along with its proof, in Appendix A.

**LLM-Check** proposes to detect hallucinations at the level of *entire responses*: for a chosen LLM layer $l$, `LLM-Chk-l` obtains an "Attention Score" $c^l$ by averaging the log-determinants of attention matrices across the set of corresponding heads (Sriramanan et al., 2024). Given the peculiar lower-triangular structure of these matrices, such scores can be calculated as:

$$c^l = \frac{1}{H} \sum_{h=0}^{H-1} \sum_{i=0}^{n-1} \log(\alpha_{i,i}^{l,h}), \tag{4}$$

i.e., by summing the log-transformed attentions paid by each token to itself, averaged across heads. In practice, the inner summation is replaced with averaging, more robust to prompt and response lengths. We note, in our CHARM, these "self-scores" are gathered and processed as node features, which can be transformed and then later aggregated by our architecture to reproduce scores $c^l$'s.

**Proposition (informal) 2.** *With a single-layer message-passing stack $f_{mp}$, CHARM can arbitrarily well approximate global `LLM-Chk-l` features $c^l$, provided attentions are clipped away from zero[3].*

*Proof idea.* Intuitively: (1) up MLPs can calculate the initial log-transform on the features of the receiving node/token $i$ — that is, on each token's "self-scores" — while discarding information from neighbours; (2) $f_{pool}$, set to summation — or averaging, if required — can aggregate these values across tokens; (3) last, $f_{pred}$ can average these values across heads and selectively for the desired layer $l$ to implement the outer averaging in Equation (4). Again, a formal statement of informal proposition 2 is reported and proved in Appendix A.

Both the two above propositions guarantee that, while general and learnable, our approach can also provably default to known, hand-crafted heuristics (under mild, reasonable assumptions). This showcases the expressiveness of the CHARM framework, further evidencing how graph representations and message-passing networks can offer a valid and compelling perspective into the task of HD.

## 5 EXPERIMENTS

We evaluate different aspects of learning with CHARM through the following research questions: **(Q1)** Is our GNN-based formulation effective in practice? **(Q2)** Is CHARM effective at detecting different

---

[3]This assumption ensures the log is continuous on an appropriate compact set, rendering its approximation amenable (see Appendix A); in practice we also did observe it was necessary to ensure numerical stability.

types of hallucinations and across different granularities (e.g., token-level and full-response)? **(Q3)** Is the underlying graph structure crucially contributing for CHARM's performance? **(Q4)** Can CHARM handle large/dense graphs? **(Q5)** Can the combination of attention and activations be effective in CHARM? **(Q6)** Does CHARM exhibit any zero-shot transferability across datasets?

We initialise training of CHARM with 3 different random seeds, and, in the following, report the mean test performance along with std. All results are obtained by models optimising validation performance (AUPR, see below). Additional information, including dataset details, hyperparameter searches, implementation notes, and extended results, are available in Appendices B to D[4]. Last, we refer readers to Appendix E for sample visualisations of the constructed computational trace graphs in input to CHARM along with token-wise hallucination predictions.

## 5.1 CONTEXTUAL TOKEN-LEVEL HALLUCINATION DETECTION

**Datasets.** We first evaluate our approach at a token-level granularity on the NQ (Kwiatkowski et al., 2019) and CNN (See et al., 2017) datasets. These consists of prompt-response pairs with hallucination annotations available at the level of single response tokens (see Section 4.1). These pairs are obtained by prompting a target LLM to perform either document-based question answering (NQ) or text summarisation (CNN). These datasets contain instances of *contextual hallucinations*: although the relevant and correct facts are provided in the input context, the target LLM is still observed to generate incorrect responses (Chuang et al., 2024). Original generations and annotations for this dataset are derived from (Chuang et al., 2024)[5]; coherently with the setup in the same work, we take `LLaMa-2-7B-chat` as the reference LLM on both datasets. More details are in Appendix B.1.

**Method comparisons.** We compare against a set of representative baselines. *Probability-based detectors* (`Probas`) (Guerreiro et al., 2023; Kadavath et al., 2022; Varshney et al., 2023; Huang et al., 2025) leverage the next-token probabilities to estimate LLM uncertainty and predict hallucinations; *Activation probes* (Orgad et al., 2025; Azaria & Mitchell, 2023; Belinkov, 2022) (`Act-*`) train a logistic classifier on *activations* at specific layers; the *attention-based*, `Lookback Lens` heuristic (Chuang et al., 2024) fits the same model on hand-crafted token-wise features calculated over all layers and heads (see Equation (3)).

Table 1: Test AUROC and AUPR (%) for NQ and CNN (token-wise, higher is better). **Bold**: best, Underlined: runner-up. [†]: we also *tune* the regularisation strength, differently than in the original work.

| Method | NQ | | CNN | |
|---|---|---|---|---|
| | AUROC | AUPR | AUROC | AUPR |
| Probas | 49.8 | 16.2 | 54.4 | 8.2 |
| Act-24 | 73.0 | 36.2 | 71.3 | 20.3 |
| Act-28 | 71.6 | 34.6 | 70.1 | 18.4 |
| Act-32 | 67.4 | 28.6 | 67.7 | 15.4 |
| Lookback Lens | 70.8 | 31.0 | 71.9 | 17.4 |
| Lookback Lens [†] | 71.9 | 34.3 | 74.4 | 19.7 |
| Neigh-Avg(N) | 66.0 | 24.5 | 70.1 | 14.9 |
| Neigh-Avg(E) | 66.8 | 30.4 | 70.5 | 18.6 |
| CHARM (att) | **74.8**±0.6 | **40.3**±1.7 | **75.4**±0.2 | **22.7**±0.4 |
| CHARM (att+act-24) | 72.2±1.2 | 35.5±1.6 | 70.9±0.2 | 19.8±0.5 |

(ours)

We run `Act-*`'s on the common choice of LLM layers 24, 28, 32, motivated by the findings in (Chuang et al., 2024; Azaria & Mitchell, 2023). As for CHARM, we instantiate it in two configurations: one only employing attention features (att), another also utilising activations from a specific layer, which we set to 24 (att+act-24) due to its consistently superior performance in `Act-*`'s We run CHARM on graphs sparsified with a fixed $\tau = 0.05$ (see Equation (1) and related ablations in Section 5.3). The HD task is at the level of single nodes/tokens, so $f_{pool}$ is set to *identity*. We additionally consider two ablated versions of CHARM (att): `Neigh-Avg(N)`, `Neigh-Avg(E)`. These extract token-wise features through a single, non-learnable msg-passing step, aggregating, resp., either node or edge features across neighbourhoods in the same computational trace graphs considered in CHARM (details are in Appendix D.1.1). Comparing with these allows us to evaluate the relevance of our multi-layer, learnable procedure. *All methods* in comparison have their hyperparameters tuned on the Val. set. Performance is measured in terms of Test AUROC and AUPR.

---

[4]Code available at `https://github.com/Noired/charm`.

[5]Differently than (Chuang et al., 2024), however, we construct and experiment with a different split ensuring full textual disjointness between train, test, and validation, see more in Appendix B.1.

**Results,** reported in Table 1, show our approach consistently outperforms all baselines across both datasets and metrics in the att.-only configuration. The significant margins over `Lookback Lens`, `Neigh-Avg(N)` and `Neigh-Avg(E)` underscore the benefits of an expressive, learnable graph-based method over attention-based heuristics. This pure attention configuration also surpasses *all* activation-based `Act-*` probes, this contributing to answer positively to **Q1**. We interestingly report that including activations from an intermediate layer into `CHARM` reveals detrimental. We hypothesise that, on these contextual benchmarks, attention features alone carry most of the relevant predictive signal — an expressive enough model like ours can leverage it at best and struggle to find additional complementary signals on activations, which could, instead lead to fit spurious correlations.

## 5.2 RESPONSE-LEVEL HALLUCINATION DETECTION

**Datasets.** We next evaluate `CHARM` at the coarser response-level HD on three datasets: Movies (Orgad et al., 2025), WinoBias (Zhao et al., 2018), and Math (Sun et al., 2024). Unlike NQ and CNN, these benchmarks address failure modes different than contextual grounding, namely: factual knowledge recall (Movies), intrinsic bias in coreference resolution (WinoBias), and arithmetic reasoning (Math). This allows to assess generalisation across fundamentally different hallucination types. The relative role of attention thereon is not obviously clear, but we hypothesise they can provide informative signals, e.g. by capturing systematic biases in attention to demographic cues or by reflecting unusual patterns in intermediate calculations steps. Exploring their interplay with other computational traces is thus a insightful direction. For these experiments we derive text generations and hallucination annotations following the procedure in (Orgad et al., 2025) and, consistently with this work, we target a different LLM, `Mistral-7B-instruct`. More dataset details are in Appendix B.2.

**Method comparisons.** As for `CHARM` and its non-learnable counterparts (iv), we set component $f_{\text{pool}}$ to *average*. We compare our method to `Act-*`'s probing activations in notoriously relevant token positions, e.g., the last token of the prompt, or the last of the response (Orgad et al., 2025) (see Appendix C.4). Here, we compare against the response-level attention-based LLM-Check (Sriramanan et al., 2024) (`LLM-Chk-*`) and the spectral-method proposed in (Binkowski et al., 2025) (`LapEig`). We also run an enhanced counterpart of (`LLM-Chk++-*`) whereby per-head scores are considered as inputs to logistic regression, rather than being averaged (Equation (4)).

**Results** are reported in Table 2. Overall, `CHARM` attains the best performance across these benchmarks, with particularly notable gaps in Math. Together with the above, these results answer positively to **Q1** and **Q2**. Interestingly, we observe a markedly different behaviour than in the previously considered contextual HD datasets. Other than Movies — where both our configurations work equally well — on

Table 2: Test AUROC and AUPR (%) for Movies, Winobias, Math (response-lv, higher is better). **Bold**: best, Underlined: runner-up.

| Method | Movies | | Winobias | | Math | |
|---|---|---|---|---|---|---|
| | AUROC | AUPR | AUROC | AUPR | AUROC | AUPR |
| Probas | 58.6 | 81.6 | 64.5 | 20.0 | 54.5 | 57.4 |
| Act-24 | 77.0 | 90.4 | 76.6 | 37.8 | 77.7 | 77.5 |
| Act-28 | 77.0 | 90.4 | 73.9 | 34.3 | 78.1 | 77.8 |
| Act-32 | 76.3 | 90.2 | 72.7 | 35.3 | 76.6 | 77.9 |
| LLM-Chk-24 | 47.5 | 74.6 | 38.9 | 10.9 | 64.5 | 68.2 |
| LLM-Chk-28 | 51.1 | 76.7 | 41.6 | 11.4 | 65.5 | 69.2 |
| LLM-Chk-32 | 61.5 | 82.1 | 41.6 | 11.3 | 64.0 | 67.6 |
| LLM-Chk++-24 | 66.3 | 84.5 | 64.6 | 20.5 | 67.3 | 69.1 |
| LLM-Chk++-28 | 67.8 | 86.3 | 64.8 | 21.0 | 67.0 | 70.6 |
| LLM-Chk++-32 | 73.0 | 88.8 | 67.2 | 24.1 | 68.6 | 72.5 |
| LapEig | 72.9 | 88.4 | 74.1 | 33.3 | 73.6 | 76.3 |
| Neigh-Avg(N) | 78.6 | 91.2 | 63.8 | 23.0 | 77.4 | 79.2 |
| Neigh-Avg(E) | 54.9 | 78.5 | 65.8 | 21.9 | 76.7 | 78.3 |
| CHARM (att) | **80.3**±0.2 | **92.0**±0.1 | 70.4±0.7 | 29.1±1.0 | 76.5±1.1 | 79.7±0.5 |
| CHARM (att+act-24) | 79.7±0.3 | 91.8±0.2 | **77.8**±0.4 | **39.8**±1.3 | **80.8**±0.7 | **83.1**±0.7 |

both Winobias and Math, activation-based features work in synergy with attention-based ones. Instead of leading to fit spurious correlations as *hypothesised* for NQ and CNN (Section 5.1), they contribute to strongly boost performance over the att.-only `CHARM`, and over all considered att.- and act.-based methods. This is particularly pronounced on Winobias — there, `CHARM` (att.) is surpassed by `Act-*`'s, but the inclusion of activations leads it (att. & act.-24) to significantly outperform them

both. Overall these datasets provide cases leading to a positive answer to **Q5**. We last note that Math is the only dataset where `Act − 24` outperformed by other variants, namely `Act-32`. We thus also ran CHARM (att+act-32), which scored Test AUROC of $81.7 \pm 0.2$, and AUPR of $83.8 \pm 0.3$.

## 5.3 Additional Analyses

We report here a series of additional analyses investigating: (i) The contribution of the graph structure to CHARM's performance; (ii) The impact of the sparsification threshold $\tau$ to the computational efficiency and performance of CHARM; (iii) The zero-shot cross-dataset transfer capabilities of CHARM and comparable approaches. Additional experimental studies are reported in Appendix C, including analyses on the impact of the activation layer choice (Appendix C.2) and on the generalisation performance CHARM on sequences of unseen length (Appendix C.3).

**The contribution of the graph structure.** To what extent does message-passing on the attention-induced graph contribute to the performance of CHARM? To answer this, we ablate the connectivity in our input samples and train CHARM on two representative datasets: CNN and Math. In this setup,

Table 3: Results from ablating the graph structure.

| Method | CNN | | Math | |
|---|---|---|---|---|
| | AUROC | AUPR | AUROC | AUPR |
| CHARM (no g.) | 70.8 $\pm0.5$ | 19.2 $\pm0.5$ | 80.6 $\pm0.7$ | 82.7 $\pm0.1$ |
| CHARM | **75.4** $\pm0.2$ | **22.7** $\pm0.4$ | **81.7** $\pm0.2$ | **83.8** $\pm0.3$ |

CHARM defaults to a set model which, instead of message-passing, applies a stack of dense layers over node features. We train and extensively tune this baseline, denoted "CHARM (no g.)", considering both att.-only and att. & act. configurations. We report its best results in Table 3 compared to the best corresponding CHARM. Results clearly show the positive contribution of message-passing on the constructed topology, answering positively to **Q3**.

**The impact of $\tau$ on efficiency and performance.** We experiment with different values of the attention threshold $\tau$ (Equation (1)), studying how graph sparsity and (inference) memory consumption vary in relation to performance. We run this study on NQ, with results in Table 4. Test AUPR is reported along with the number of edges, sparsity and inference memory footprint averaged over test graphs. We observe

Table 4: *Avg.* graph stats. on NQ at different thresholds, along with GPU memory footprint and Test AUPR.

| $\tau$ | Num. Edges | Sparsity | Mem. (MB) | AUPR |
|---|---|---|---|---|
| 0.5 | 1,118.80 | 0.993 | 22.99$\pm3.71$ | 38.4$\pm0.4$ |
| 0.1 | 7,458.67 | 0.952 | 60.44$\pm11.79$ | 41.0$\pm1.2$ |
| 0.05 | 14,884.44 | 0.906 | 104.15$\pm23.05$ | 40.3$\pm1.7$ |
| 0.01 | 58,998.88 | 0.645 | 363.02$\pm98.39$ | 40.3$\pm0.9$ |
| 0.001 | 19,7784.82 | 0.026 | 1177.20$\pm523.61$ | 40.1$\pm0.0$ |

CHARM's performance is robust to various levels of sparsifications, whilst this can provide dramatic reduction in resource consumption. Performance drops more notably only for $\tau = 0.5$, which, we note, still outperforms the best competitor, i.e., `Act-24` (see Table 1). Overall our default $\tau = 0.05$ attains a good trade-off, whilst we note it maximises val. AUPR. These results answers positively to **Q4**. We finally measure a distinctly contained inference latency of $\approx 1e^{-3}$ secs. Refer to Appendix C.1 for run-time and performance comparisons with other popular HD methods relying on multiple prompting, which incur significantly higher latency.

**Transferring zero-shot to unseen datasets.** CHARM is a learnable, expressive multi-layer approach — this raises a natural question: *To what extent can it generalise cross-datasets zero-shot?* To investigate this, we follow the setup in (Chuang et al., 2024): we train on NQ and evaluate on CNN, and vice-versa. Results are in in Table 5. Overall *no single*

Table 5: Cross-dataset zero-shot transf. NQ $\leftrightarrow$ CNN.

| Method | NQ $\rightarrow$ CNN | | CNN $\rightarrow$ NQ | |
|---|---|---|---|---|
| | AUROC | AUPR | AUROC | AUPR |
| LkbLens [†] | **68.6** | **14.9** | 62.0 | 26.5 |
| Act-24 | 63.4 | 11.3 | 63.8 | 29.7 |
| CHARM | 64.1 $\pm1.1$ | 12.0 $\pm0.98$ | **65.5** $\pm0.14$ | **31.6** $\pm0.10$ |

*method consistently outperforms the others* in this challenging setup. In fact, despite its larger expressiveness, CHARM demonstrates promising generalisation: it outperforms activation-based probes, ranks best in CNN $\rightarrow$ NQ, and places second in NQ $\rightarrow$ CNN (behind `Lookback Lens`, which conversely performs the *worst* in CNN $\rightarrow$ NQ). These results suggest that zero-shot transfer

remains an open challenge, but our graph-based formulation is competitive and can capture generalisable signals, answering positively to **Q6**.

## 6 CONCLUSIONS

In this work, we proposed attributed graphs as a principled formulation of LLM computational traces, showing how diverse signals can be unified in this framework and how neural message passing can be applied thereon for diverse HD tasks. We showed our approach, CHARM, can provably generalise prior methods and that it achieves strong empirical performance, consistently outperforming existing methods. Additional analyses underscored the importance of graph structure and demonstrated promising zero-shot generalization across datasets.

Future endeavours will consider integrating other computational traces (e.g., logits), as well as extensions to new tasks such as detecting data contamination, identifying LLM-generated text, or flagging jailbreak attempts. Future work may also explore alternative message-passing architectures, including positional and structural encodings tailored to these attributed graphs.

### ACKNOWLEDGMENTS

F.F. conducted this work supported by an Aly Kaufman Post-Doctoral Fellowship. G.B. is supported by the Jacobs Qualcomm PhD Fellowship. HM is supported by the Israel Science Foundation through a personal grant (ISF 264/23) and an equipment grant (ISF 532/23), and by the Career Advancement Chairs in Artificial Intelligence – Schmidt Futures. F.F. is extremely grateful to the members of the "Eva Project", whose support he immensely appreciates.

## REPRODUCIBILITY STATEMENT

Our code is available at `https://github.com/Noired/charm`, along with training and evaluation scripts. Section 5, along with Appendices B to D provide the required implementation details to reproduce our results.

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

# A EXPRESSIVENESS: CLAIMS AND PROOFS

**Proposition (informal) 1.** *Equipped with a single-layer message-passing stack $f_{mp}$, CHARM running on a non-tresholded computational trace graph ($\tau = 0$) can arbitrarily well approximate token-wise* Lookback Lens *features $\ell_i$ for bounded prompt and response lengths.*

**Proposition 1.** *Let $\ell_i(\vec{s}; \mathcal{L})$ denote the Lookback Lens features calculated, for token $i$ on string $\vec{s} = \vec{p} \mid \vec{r}$ for LLM $\mathcal{L}$(Equation (3)). Also, let $h_i^{(t)}(G_{\vec{s}})$ denote the $t$-layer representation for the same token in output from $t$ CHARM msg-passing layers (Equation (2)) on the corresponding computational trace graph $G_{\vec{s}}$. For any precision $\epsilon > 0$, there exists a 1-layer stack of CHARM's layers which approximate Lookback Lens features up to precision $\epsilon$, for maximum allowed prompt-length $\bar{n}_p$ and response length $\bar{n}_r$. That is, for any prompt $\vec{p}$, response $\vec{r}$ of the aforementioned maximum lengths, and for any response token $i$, $\|h_i^{(1)}(G_{\vec{p}|\vec{r}}) - \ell_i(\vec{p} \mid \vec{r}; \mathcal{L})\|_\infty < \epsilon$.*

*Proof.* To prove the above we show that, by setting hyperparameters $\square \equiv \sum$ (summation) and $\tau = 0$ (no thresholding), there exist a single-layer stack $f_{\text{mp}}$ which compute the desired approximation. In the following we will consider the attention scores across the $L$ layers and $H$ heads to be arranged in our node and edge features in a flattened manner. We will conveniently denote with $\flat(l, h)$ the function which evaluates the index of layer $l$ and head $h$ in this flattened representation.

**(0)** *Setup and inputs.* As our stack is made up of one single message-passing layer, our specific interest is thus in showing the existence of appropriate MLPs $\text{msg}_0$, $\text{up}_0$ enabling $f_{\text{mp}}$ to realise the target approximation. Being these components of the *first* — and only — message-passing layer in $f_{\text{mp}}$, its input node and edge representations effectively correspond to the original node and edge features $X_V, X_E$. We are thus focussing on the following update:

$$h_i^{(1)} = \underbrace{\text{up}_0}_{(2)} \left( x_{V,i}, \sum_{j < i} \underbrace{\text{msg}_0}_{(1)} \left( x_{V,i}, x_{V,j}, x_{E,(i,j)}, p_{i,j} \right) \right) \tag{5}$$

where the neighbourhood aggregation sums across *all* previous token positions ($j < i$) since no thresholding is enforced ($\tau = 0$) and due to the fact that attention values cannot exactly evaluate to $0$ because of to the application of softmax normalisation.

**(1)** *Message function.* Let us first describe what we desire $\text{msg}_0$ to calculate. We would like it to map the concatenation of its arguments, with dimensionality $3d + 2$, $d = L \cdot H$, to a vector of dimensionality $2d + 2$, where:

- The mark feature $p_{i,j}$ is replicated in the first two dimensions (channels $0, 1$);

- Edge features $x_{E,(i,j)}$ are replicated either in channels $2$ through $2 + d - 1$ if $p_{i,j} = [1, 0]$ (message from prompt token) or in channels $2 + d$ through $2d - 1$ otherwise (message from response token);

- Node features $x_{V,i}, x_{V,j}$ are discarded.

Now, an MLP exactly implementing the above message function does exist; in fact, it can be *explicitly constructed.*

*First layer.* Weight matrix $W_1$ is in $\mathbb{R}^{(3d+2) \times (2d+2)}$. We will describe it in terms of columns slices.

- A first slice gathers the first two columns $(0, 1)$; these are all zero except for the bottom $2 \times 2$ block, set as identity $I_2$. This slice copies the $p_{i,j}$ mark features in the first two channels of the hidden representation.

- A second slice gathers columns $2$ through $2 + d - 1$; these are all zero except for rows $2d$ through $3d - 1$, set to identity $I_d$, and row $3d$, set to a $\vec{1}_d$ one-only vector. This slice calculates the sum between edge features $x_{E,(i,j)}$ and the first channel of the mark $p_{i,j}$, indicating whether the message comes from a prompt token.

- A third — and last — slice gathers columns $2 + d$ through $2 + 2d - 1$; these are all zero except for rows $2d$ through $3d - 1$, set to identity $I_d$. This slice copies edge features $x_{E,(i,j)}$ in the last $d$ channels of the hidden representation.

Bias vector $b_1$ in $\mathbb{R}^{2d+2}$ is all zero except for channels 2 through $2 + d - 1$, set to vector $-\vec{1}_d$. Recapping, the hidden representation is a vector in $\mathbb{R}^{2d+2}$ with the following structure:

$$\left[ p_{i,j} \mid \underbrace{\cdots \left(x_{E,(i,j)}\right)_{\flat(l,h)} + \left(p_{i,j}\right)_0 - 1 \cdots}_{(h_1)} \mid \underbrace{\cdots \left(x_{E,(i,j)}\right)_{\flat(l,h)} \cdots}_{(h_2)} \right].$$

*Second layer*: Weight matrix $W_2$ is in $\mathbb{R}^{(2d+2) \times (2d+2)}$. We will describe it in terms of columns slices again.

- A first slice gathers the first two columns $(0, 1)$; these are all zero except for the top $2 \times 2$ block, set as identity $I_2$. This slice replicates, again, the $\mathrm{ReLU}(p_{i,j}) = p_{i,j}$ mark features in the first two channels of the output.

- A second slice gathers columns 2 through $2 + d - 1$; these are zero in the first two rows, rows 2 through $2 + d - 1$ are set to identity$-I_d$ and the last $d$ rows are set to $I_d$. This slice calculates, channel-by-channel, $\mathrm{ReLU}(h_2) - \mathrm{ReLU}(h_1) = h_2 - \mathrm{ReLU}(h_1)$.

- A third — and last — slice gathers columns $2 + d$ through $2 + 2d - 1$; these are all zero except for rows 2 through $2 + d - 1$, set to identity $I_d$. This slice copies $\mathrm{ReLU}(h_1)$ in the last $d$ channels of the output.

Bias vector $b_2$ is set to zero.

As a result, the output is a vector in $\mathbb{R}^{2d+2}$ with the following structure:

$$\left[ p_{i,j} \mid \underbrace{\cdots \left(x_{E,(i,j)}\right)_{\flat(l,h)} - \mathrm{ReLU}(\left(x_{E,(i,j)}\right)_{\flat(l,h)} + \left(p_{i,j}\right)_0 - 1) \cdots}_{(o_1)} \mid \underbrace{\cdots \mathrm{ReLU}(\left(x_{E,(i,j)}\right)_{\flat(l,h)} + \left(p_{i,j}\right)_0 - 1) \cdots}_{(o_2)} \right].$$

Now, note that, as desired:

- if $\left(p_{i,j}\right)_0 = 1$ (message from response token):
  - $\left(o_1\right)_{\flat(l,h)} = \left(x_{E,(i,j)}\right)_{\flat(l,h)} - \mathrm{ReLU}(\left(x_{E,(i,j)}\right)_{\flat(l,h)} + 1 - 1) = \left(x_{E,(i,j)}\right)_{\flat(l,h)} - \left(x_{E,(i,j)}\right)_{\flat(l,h)} = 0$
  - $\left(o_2\right)_{\flat(l,h)} = \mathrm{ReLU}(\left(x_{E,(i,j)}\right)_{\flat(l,h)} + 1 - 1) = \left(x_{E,(i,j)}\right)_{\flat(l,h)}$,

- if $\left(p_{i,j}\right)_0 = 0$ (message from prompt token):
  - $\left(o_1\right)_{\flat(l,h)} = \left(x_{E,(i,j)}\right)_{\flat(l,h)} - \mathrm{ReLU}(\left(x_{E,(i,j)}\right)_{\flat(l,h)} + 0 - 1) = \left(x_{E,(i,j)}\right)_{\flat(l,h)} - 0 = \left(x_{E,(i,j)}\right)_{\flat(l,h)}$
  - $\left(o_2\right)_{\flat(l,h)} = \mathrm{ReLU}(\left(x_{E,(i,j)}\right)_{\flat(l,h)} + 0 - 1) = 0$

Now, when aggregating these calculated messages through summation, it becomes clear that the aggregated message vector $m_i$ will eventually hosts:

- The number of response tokens preceding token $i$, i.e., $i - n_p - 1$, in channel 0;

- The length of the prompt, i.e., $n_p$, in channel 1;

- Summation $\sum_{j=0}^{n_p-1} \alpha_{l,h}^{i,j}$ in channel $\flat(l, h) + 2$, denoted $\hat{P}_i^{l,h}$;

- Summation $\sum_{j=n_p}^{i-2} \alpha_{l,h}^{i,j}$ in channel $\flat(l, h) + 2 + d$, denoted $\left(\hat{R}_i^{l,h}\right)^-$.

**(2) *Update function.*** Next, we desire $\mathrm{up}_0$ to implement a function $\mathrm{up}^*$ mapping the concatenation $\begin{bmatrix} x_{V,i} \mid m_i \end{bmatrix}$ to a vector of dimension $d$ corresponding to the `Lookback Lens` output scores in Equation (3). Showing that $\mathrm{up}_0$ can approximate $\mathrm{up}^*$ up to desired precision $\epsilon$ will complete the proof.

We now describe $\mathrm{up}^*$. We build it as a composition of two functions $f_A \circ f_B$:

- $f_A$ is such that $f_A(x_{V,i}, m_i) = y_i = c_i + m_i$, where $c_i$ has the same dimensionality as $m_i$ $(2d + 2)$, and $c_i = \begin{bmatrix} 0 \mid 1 \mid \vec{0} \mid x_{V,i} \end{bmatrix}$ — note that vector $y_i$ is an "updated" version of $m_i$ whereby channel 1 now equals $i - n_p$ and channels $\flat(l, h) + 2 + d$'s are now such that:

$$\left(\hat{R}_i^{l,h}\right)^- + x_{V,i} = \sum_{j=n_p}^{i-1} \alpha_{l,h}^{i,j} = \hat{R}_i^{l,h} \tag{6}$$

- $f_B$ is such that $f_B(y_i) = z_i$ where $z_i$ is of dimensionality $d$ and:

$$\left(z_i\right)_{\flat(l,h)} = \frac{\dfrac{\left(y_i\right)_{\flat(l,h)+2}}{\left(y_i\right)_1}}{\dfrac{\left(y_i\right)_{\flat(l,h)+2}}{\left(y_i\right)_1} + \dfrac{\left(y_i\right)_{\flat(l,h)+2+d}}{\left(y_i\right)_0}}. \tag{7}$$

Note that, importantly, $\left(z_i\right)_{\flat(l,h)} = \dfrac{\hat{P}_i^{l,h}/n_p}{\hat{P}_i^{l,h}/n_p + \hat{R}_i^{l,h}/(i-n_p)} = \dfrac{P_i^{l,h}}{P_i^{l,h} + R_i^{l,h}} = \ell_i^{l,h}$.

First, $f_A$ can be realised by a single affine transformation. This has weight matrix $W_A$ in $\mathbb{R}^{(3d+2) \times (2d+2)}$, described in terms of row slices as follows.

- A first slice gathers the first $d$ rows; it is zero except for the last $d$-column block, set as identity $I_d$.

- A second slice gathers rows $d$ through $3d + 2 - 1$ and it set as an identity $I_{(2+2d)}$.

The above linear transformation has the effect of copying $m_i$ in the output and of summing $x_{V,i}$ in its last $d$ entries — where the aggregated message from response tokens is stored. Now, bias vector $b_A$ is in $\mathbb{R}^{2+2d}$ and is zero everywhere except for its first element which is set as 1. Adding $b_A$ has the effect of simply increasing the second entry by one, thus "updating" the count of response messages stored there.

Second, we note that $f_B$ computes the same exact scalar-valued function $f_B^s$ on each output component; also, this $f_B^s$ is continuous on (the non-compact) domain $\{1, \ldots, \bar{n}_p\} \times \{1, \ldots, \bar{n}_r\} \times (0, 1)^2$. We note that $f_B^s$ can be trivially, continuously extended to the compact $\{1, \ldots, \bar{n}_p\} \times \{1, \ldots, \bar{n}_r\} \times [0, 1]^2$: it suffices to see that its limits exist and are finite on the boundary of $[0, 1]^2$. This is indeed the case; we note that denominators in each individual normalisation ratios are always greater or equal than one; and that the two addenda in the denominator of the main ratio are always non-negative, can never evaluate simultaneously to zero and their sum is bounded away from zero (given the maximum allowed length for prompt and response).

Given this premise, term $f_B^{s,\mathrm{ext}}$ this continuous extension; we can invoke the MLP Universal Approximation Theorem (Pinkus, 1999) to claim the existence of an MLP $M_B^s$ with one hidden layer which approximates the continuous $f_B^{s,\mathrm{ext}}$ on the compact domain $\{1, \ldots, \bar{n}_p\} \times \{1, \ldots, \bar{n}_r\} \times [0, 1]^2$ up to precision $\epsilon$. This implies the original $f_B^s$ is also $\epsilon$-approximated in its original domain. Now, it is possible to (easily,) appropriately *replicate* the weights of $M_B^s$ to distribute its same exact computation for each of the output coordinates, thus obtaining an MLP $M_B$ approximating the overall $f_B$: $\forall y, \forall i \, |M_B^s(y) - \left(f_B(y)\right)_i| < \epsilon$, that is, $\forall y, \forall i \, |\left(M_B(y)\right)_i - \left(f_B(y)\right)_i| < \epsilon$. But, then:

$$\forall y, \forall i \, |\left(M_B(y)\right)_i - \left(f_B(y)\right)_i| < \epsilon$$
$$\implies \forall y \, \max\left(|\left(M_B(y)\right)_i - \left(f_B(y)\right)_i| \,\Big|\, i = 0, \ldots, d-1\right) < \epsilon$$
$$\implies \forall y \, \|M_B(y) - f_B(y)\|_\infty < \epsilon$$

Denote $(W_B^1, b_B^1), (W_B^2, b_B^2)$ the weight and biases of, respectively, the first and second layers of $M_B$. Now, the above affine transformation exactly implementing $f_A$ can be "absorbed" into the *first* layer of $M_B$, by replacing the weight and bias as $W^1 = W_B^1 \cdot W_A, b_1 = (W_B^1 \cdot b_A + b_B^1)$. The resulting MLP composed by $\left((W^1, b^1), (W_B^2, b_B^2)\right)$ now $\epsilon$-approximates the overall up*, concluding the proof.

$\square$

**Proposition (informal) 2.** *With a single-layer message-passing stack $f_{mp}$, CHARM can arbitrarily well approximate global* LLM-Chk-l *features $c^l$, provided attentions are clipped away from zero[6].*

**Proposition 2.** *Let $c^l(\vec{s}; \mathcal{L})$ denote the LLM-Check Attention Score (Equation* (4)*) calculated on string $\vec{s} = \vec{p} \mid \vec{r}$ for LLM $\mathcal{L}$(Equation* (4)*) and its layer $l$. Also, let $y(G_{\vec{s}})$ denote the prediction in output from an CHARM stacking components $f_{msg}, f_{pool}, f_{pred}$, run on corresponding graph $G_{\vec{s}}$. For any precision $\epsilon > 0$, when $f_{pool} \equiv \sum$ and attention values are clipped from below to value $\alpha_{min} > 0$, there exists a 1-layered $f_{msg}$, and an MLP $f_{pred}$ such that CHARM approximates LLM-Check Attention Scores up to precision $\epsilon$. That is, for any prompt-response pair $\vec{s}$, $|y(G_{\vec{s}}) - c^l(\vec{s}; \mathcal{L})| < \epsilon$.*

*Proof.* To prove the above we show that, in the setting described in the proposition, there exist a single-layer stack $f_{mp}$, as well as an MLP $f_{pred}$ which compute the desired approximation. Here we consider the LLM-Chk-l variant which averages across heads instead of performing a summation, but the proof below is easily extended to this alternative configuration.

**(0) *Setup, inputs and proof strategy.*** Our stack is made up, again, of one single message-passing layer in $f_{mp}$, followed by sum-based pooling and an MLP $f_{pred}$. Our specific interest is in showing the existence of appropriate MLPs $msg_0$, $up_0$, $f_{pred}$ enabling the full stack to realise the target approximation. Again, $msg_0$, $up_0$ are the components of the *first* — and only — message-passing layer in $f_{mp}$, so its input node and edge representations effectively correspond to the original node and edge features $X_V, X_E$. The whole computation then takes the form:

$$y = \underbrace{f_{pred}}_{(3)} \Big( \sum_{i=0}^{n-1} \underbrace{up_0}_{(2)} \Big( x_{V,i}, \sum_{(i,j) \in E} \underbrace{msg_0}_{(1)} \big(x_{V,i}, x_{V,j}, x_{E,(i,j)}, p_{i,j}\big) \Big) \Big) \tag{8}$$

**(1) *Message function.*** The LLM-Chk-l method does not perform any aggregation on the attention graph — for our purposes it suffices for MLP $msg_0$ to simply implement a function which outputs any constant non-negative vector $v$. W.l.o.g., set $v = \vec{0}$; the MLP implementing $msg_0$ is trivially obtained by setting both its weights and biases to zero.

**(2) *Update function.*** Note that $x_{V,i} = \alpha_{i,i}$; as it will be clearer next, it suffices for $up_0$ to approximate the $\log$ function applied thereon component-wise. Now, $\log$ is, in fact, operating on domain $[\alpha_{min}, 1)$ in view of the applied clipping; there, the function is continuous. Consider the compact set $[\alpha_{min}, 1]$ obtained as the closure of the above domain. The considered $\log$ is trivially continuously extended to this new domain, since its limit exists finite as the argument approaches $1$ from the left (it evaluates to $0$). We therefore invoke the Universal Approximation Theorem (Pinkus, 1999), which guarantees the existence of an MLP $M_{\log}$ approximating the component-wise $\log$ function on the extended domain and, in turn, on the original input domain $[\alpha_{min}, 1)$ up to arbitrary precision $\epsilon$. We construct $up_0$ by appropriately replicating the weights of $M_{\log}$ to output the approximated log-transform on each of the first input $d$ channels in parallel, whilst discarding the last $d$ remaining channels in output from the message function (see above).

**(3) *Prediction function.*** At this point, the input of $f_{pred}$ corresponds to: $\hat{z} = \frac{1}{n} \sum_{i=0}^{n-1} \hat{y}_i$, where $\hat{y}_i$ is an $\epsilon$-approximation of the $\log$ function applied element-wise to $\alpha_{i,i}$'s. If $f_{pred}$ implemented the final averaging over $l$'s $H$ heads (channels), then its output would be an overall approximation of the desired quantity $c^l$ (of a certain precision yet to be quantified). We note that it easy to explicitly construct such an MLP exactly implementing the outer averaging over the selected heads. We describe its two layers in the following.

---

[6]This assumption ensures the $\log$ is continuous on an appropriate compact set, rendering its approximation amenable (see Appendix A); in practice we also did observe it was necessary to ensure numerical stability.

- First layer: it features a weight matrix of the form $[I_d \mid -I_d]$ and a zero-valued bias; this layer expands the $d$-dimensional input to a $2d$ representation, where the the first $d$ channels host input $\hat{z}$, the last $d$ channels the negated input $-\hat{z}$.

- Second layer: it features a weight matrix in $\mathbb{R}^{2d \times 1}$. The upper $d \times 1$ block hosts a vector where entries $\flat(l, h)$ equal to $1/H$ if $l = \bar{l}$, 0 otherwise. The lower $d \times 1$ block has the same exact structure, but non-zero entries are, instead, set to $-1/H$.

It is easy to see this construction exactly implements the required averaging, making the ReLU activaction act neutrally.

Now, we ask to what precision does the final output approximate overall target $c^l$. Note that the only source of approximation is in the previously introduced $\mathrm{up}_0$; we are thus only required to quantify how it "propagates" to the rest of the following computation. We have, by the triangular inequality:

$$
\left| \frac{1}{H} \sum_{h=0}^{H-1} \frac{1}{n} \sum_{i=0}^{n-1} (\hat{y}_i)_{\flat(l,h)} - \frac{1}{H} \sum_{h=0}^{H-1} \frac{1}{n} \sum_{i=0}^{n-1} \log(\alpha_{i,i}^{l,h}) \right| \leq
$$

$$
\frac{1}{H} \sum_{h=0}^{H-1} \left| \frac{1}{n} \sum_{i=0}^{n-1} (\hat{y}_i)_{\flat(l,h)} - \frac{1}{n} \sum_{i=0}^{n-1} \log(\alpha_{i,i}^{l,h}) \right| \leq
$$

$$
\frac{1}{H} \sum_{h=0}^{H-1} \frac{1}{n} \sum_{i=0}^{n-1} \left| (\hat{y}_i)_{\flat(l,h)} - \log(\alpha_{i,i}^{l,h}) \right| \leq
$$

$$
\frac{1}{H} \sum_{h=0}^{H-1} \frac{n\epsilon}{n} = \frac{Hn\epsilon}{Hn} = \epsilon
$$

which concludes the proof. $\qquad\square$

## B DATASET DETAILS

### B.1 NQ AND CNN

**Dataset construction.** These datasets are constructed precisely following the implementation described in (Chuang et al., 2024) and provided as part as a supplementary codebase at `https://github.com/voidism/Lookback-Lens`. From this repository we derive both prompts and pre-computed, annotated generations, which we re-use via teacher-forcing to hook out the required computational traces, namely attention and activation matrices. We tested the fidelity of these generated scores in early experiments: we recalculated original `Lookback Lens` features using the generated data and managed to reproduce the original results in (Chuang et al., 2024).

**Dataset details**, including a description of the text generation and annotation process, are found in (Chuang et al., 2024, Appendix A) and (Chuang et al., 2024, Appendix C.2), to which we refer the interested reader.

**Splitting.** Chuang et al. (2024) originally split the data randomly and in a way that, potentially, passages from the same response could appear across training and evaluation splits. We argue this is an undesired side-effect and, in an effort to cast the HD in a more challenging setup, we instead split the data at the level whole prompt-response pairs (graphs according to our framework). Specifically, we fix the seed to 42 and randomly obtain a prompt-response level split in the proportion 60% / 20% / 20% (train / val / test).

### B.2 MOVIES, WINOBIAS, MATH

**Dataset construction.** These datasets are constructed following the process described in (Orgad et al., 2025), and by leveraging the authors' code open-sourced at `https://github.com/technion-cs-nlp/LLMsKnow` (MIT License). The prompts and ground truth labels of all the three considered dataset, in particular, are provided by the authors themselves in the above codebase.

As for hallucination labels, we run the annotation process whose code is provided therein. These annotation routines are mostly based on string matching procedures.

**Dataset details,** including a description of the datasets and how prompts have been derived are provided in (Orgad et al., 2025, Appendix A.3), to which we refer the interested reader.

**Splitting.** We use the same train/test splits provided by Orgad et al. (2025), and additionally carve out a random sample of 20% of training data points, treated as our validation set. We perform this sampling by setting random seed to $42$.

## C  EXTENDED EXPERIMENTAL SECTION

### C.1  COMPARISON WITH SELF-CHECK AND MULTIPLE-PROMPTING-BASED METHODS

In this section, we compare against baseline methods that rely on additional prompting, specifically P(True) Kadavath et al. (2022) and Semantic Entropy (SE) Kuhn et al. (2023). Both approaches operate over multiple LLM generations or prompts, which introduces a non-negligible computational overhead and may hinder their applicability in real-time settings. Table 6 reports results on the Movies dataset using `Mistral-7B-instruct`. For SE, we follow the original evaluation setup (Kuhn et al., 2023), employing the DeBERTa entailment model as described in the referenced work.

Table 6: Comparison with methods relying on multiple prompting.

| Method | Mis-7B – Movies (AUC) |
| --- | --- |
| P(True) | 62.00 |
| Semantic Entropy | 70.06 |
| CHARM (att) (ours) | **80.3**$_{\pm \textbf{0.2}}$ |
| CHARM (att+act-24) (ours) | 79.7$_{\pm 0.3}$ |

We observe that in all cases, CHARM variants substantially outperform the competing approaches. To quantify the computational burden of these baselines, we measured the average runtime of SE for producing a prediction. This process involves generating 10 additional responses and clustering them by computing mutual entailments with an auxiliary DeBERTa model. On average, SE required $5.9 \pm 1.7$ seconds per evaluation. We minimised the overhead of auxiliary generations by running them in parallel through batching. Nevertheless, the clustering step alone accounts for about $1.35$ seconds of runtime, which is not negligible. These findings highlight the advantage of our method, which not only achieves higher accuracy but also operates orders of magnitude faster, with detection runtimes on this dataset in the range of $10^{-4}$ seconds.

### C.2  IMPACT OF THE CHOICE OF ACTIVATION LAYER(S)

Here we report results on the Math dataset in an effort to assess the sensitivity of CHARM across different choices of the activation layer(s). We train (and tune) CHARM with features from the layers considered for activation probes, namely $24, 28, 32$. We also experiment with considering all these layers jointly, with activations concatenated together across layers. Results are in Table 7.

CHARM remains robust w.r.t. the chosen activation layer and we observe that concatenating multiple layers together may further improve performance.

### C.3  PERFORMANCE BY LENGTH

We run additional experiments on the CNN dataset to better study how model performance, inference run-time and memory footprint vary as a function of the processed sequence lengths.

**Runtime and memory consumption by length** have their trends reported in Figure 3. These results show extremely contained run-times and memory consumptions even for lengths in the order

Table 7: Performance of CHARM with different choices of activation layers.

| Method | Math | |
|---|---|---|
| | AUROC | AUPR |
| Act-24 | 77.7 | 77.5 |
| Act-28 | 78.1 | 77.8 |
| Act-32 | 76.6 | 77.9 |
| CHARM (att) | 76.5 $\pm$1.1 | 79.7 $\pm$0.5 |
| CHARM (att+act-24) | 80.8 $\pm$0.7 | 83.1 $\pm$0.7 |
| CHARM (att+act-28) | 81.2 $\pm$1.0 | 83.4 $\pm$1.3 |
| CHARM (att+act-32) | 81.7 $\pm$0.2 | 83.8 $\pm$0.3 |
| CHARM (att+act-(24,28,32)) | **82.7** $\pm$0.1 | **84.0** $\pm$0.5 |

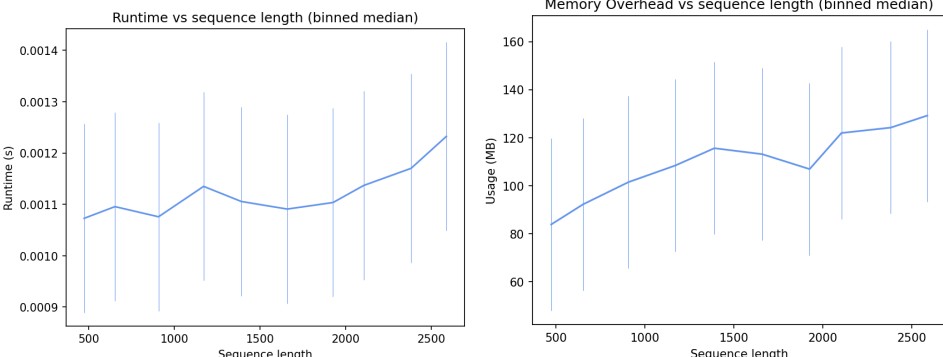

Figure 3: Inference runtime (left) and memory consumption (right) by sequence length on the CNN dataset. Values are grouped in 10 bins; x-ticks report median length per bin; the y-axis reports median measurements per bin, as well as the corresponding inter-quartile range.

of thousands of tokens. Overall, runtime and memory consumption scale very favourably in the considered range, underscoring the crucial computational advantage of running neural message-passing on *sparsified* attention graphs.

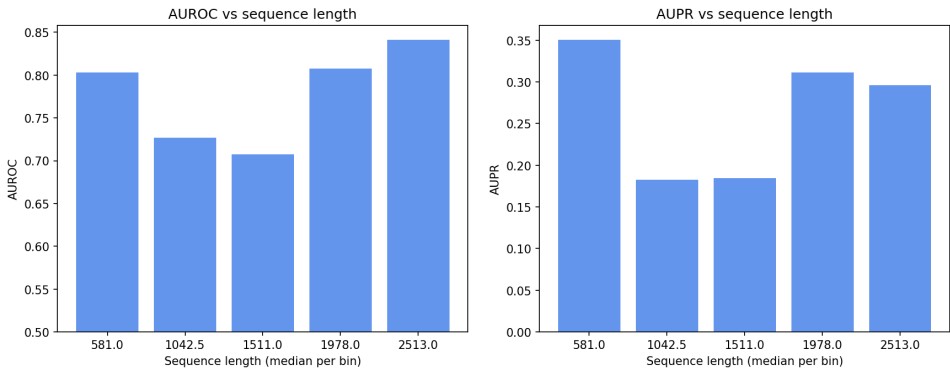

Figure 4: Test performance by sequence length on the CNN dataset: AUROC (left), AUPR (right). Values are grouped in 5 bins; x-ticks report median length per bin; the y-axis reports the performance within each bin.

**Performance by length**  is reported in Figure 4 both in terms of AUROC and AUPR. We observe that CHARM's performance does not degrade with increasing sequence length; in fact, the highest AUROC is achieved in the longest-sequence bin. Overall, these results suggest that CHARM is not particularly sensitive to sequence length.

**Generalising to longer sequences at test time.** We also experimented with training on shorter sequences and testing on longer ones, again on the CNN dataset. We produced a new non-uniform split whereby test samples are taken as the top $20\%$ longest sequences, the rest considered for training and validation sets. This way, training and model selection is run on sequences only up to $\sim 1.2$k tokens, while the model is tested on sequences of length ranging from this value to $\sim 2.7$k tokens. Results are reported in Table 8, comparing `CHARM` with other `Act-*` and `Lookback Lens` baselines.

Table 8: Performance of `CHARM` on CNN, split by size (test set contains unseen longer sequences).

| Method | CNN (longer seqs.) | |
|---|---|---|
| | AUROC | AUPR |
| `Act`-24 | 71.6 | 16.3 |
| `Act`-28 | 70.3 | 16.0 |
| `Act`-32 | 68.4 | 14.1 |
| `Lookback Lens` | 73.9 | 18.9 |
| `CHARM` (att) | **74.5** ±0.7 | **21.6** ±1.2 |

We observe that CHARM still outperforms methods in comparisons on both metrics, and that its performance on unseen sequence lengths remains stable, only marginally lower than what obtained on uniform splits (Table 1).

## C.4 HYPERPARAMETER GRIDS

We employed the same hyperparameter grid search across all datasets considered for `CHARM`, as summarized in Table 9. When incorporating activations into `CHARM`, we additionally searched over a separate weight decay parameter, applied only to the encoder of the activations, with candidate values $\{0.0, 0.05, 0.1\}$.

Table 9: Hyperparameter search space for `CHARM`.

| Hyperparameter | Values |
|---|---|
| Learning Rate | $\{0.001, 0.0005\}$ |
| Learning Rate Sched. | $\{\text{Reduce On Plateau, Cosine w/ Warmup}\}$ |
| Batch Size | 32 |
| Dropout | $\{0.25, 0.5\}$ |
| Hidden Dimension | $\{32, 64, 128\}$ |
| Number of Layers | $\{1, 2, 3\}$ |
| Weight Decay | $\{0.0, 0.001\}$ |
| BatchNorm | $\{\text{yes, no}\}$ |
| Residual Connections | $\{\text{yes, no}\}$ |

### C.4.1 BASELINE HYPERPARAMETER SEARCHES

All hyperparameters were selected based on validation performance and, in particular, in order to maximise the AUPR metric. The details are provided below.

**Probas.** We evaluated different readout functions — mean, max, and sum — applied to the next-token probabilities.

**Act-*.** We experimented with the following regularisation parameters for logistic regression: $C \in \{10^{-8}, 10^{-7}, 10^{-6}, 10^{-5}, 10^{-4}, 10^{-3}, 10^{-2}, 10^{-1}, 1, 10, 100, 10^5\}$. In addition, we probed token positions in: $\{-3, -2, -1, 0, 1, 2\}$.

**LLM-Chk-*.** We were required to clamp the attention scores from below using $\epsilon = 10^{-6}$ to avoid numerical errors. For `LLM-Chk++-*`, we experimented with $C \in \{10^{-8}, 10^{-7}, 10^{-6}, 10^{-5}, 10^{-4}, 10^{-3}, 10^{-2}, 10^{-1}, 1, 10, 100, 10^5\}$.

**LapEig.** We experimented with the following values of $k$: $\{4, 5, 6, 7, 8, 9, 10, 11, 12, 15, 20\}$. For datasets where the minimum number of tokens in the test split was less than $k$, we restricted experiments to values of $k$ below this threshold. As for logistic regression, we used $C = 1$, class balancing, and a maximum of $2,000$ iterations, consistent with what prescribed in the original paper (Sriramanan et al., 2024).

**Neigh-Avg(N) and Neigh-Avg(E).** We used mean readout exactly as in our model and tuned the logistic regression regularisation parameter over $C \in \{10^{-8}, 10^{-7}, 10^{-6}, 10^{-5}, 10^{-4}, 10^{-3}, 10^{-2}, 10^{-1}, 1, 10, 100, 10^5\}$.

**Lookback Lens.** We implemented `Lookback Lens` exactly as described in the original paper, using logistic regression with a regularization parameter of $C = 1$. For `Lookback Lens` $^\dagger$, we performed a grid search over $C \in \{10^{-8}, 10^{-7}, 10^{-6}, 10^{-5}, 10^{-4}, 10^{-3}, 10^{-2}, 10^{-1}, 1, 10, 100, 10^5\}$.

**CHARM (no g.).** We use the same exact grid as for `CHARM` (see Table 9), with the following exceptions: (1) no msg-passing layers; (2) a readout / prediction head that is either linear or a implemented an MLP.

# D  IMPLEMENTATION DETAILS AND COMPUTATIONAL RESOURCES

## D.1  DETAILED ARCHITECTURAL FORMS AND TRAINING PARAMETERS

### D.1.1  NEIGHBOURHOOD AVERAGING BASELINES

Our baselines `Neigh-Avg(N)` and `Neigh-Avg(E)` calculate features in a non-learnable way as described below.

**Neigh-Avg(N).**

$$h_i^{(1)} = \frac{1}{1 + \deg_{\text{in}}(i)} \Big( h_i^{(0)} + \sum_{j:\,(i,j)\in E} x_{E,(i,j)} \Big) =$$
$$= \frac{1}{1 + \deg_{\text{in}}(i)} \Big( x_{V,i} + \sum_{j:\,(i,j)\in E} x_{E,(i,j)} \Big) =$$
$$= \frac{1}{1 + \deg_{\text{in}}(i)} \Big( \boldsymbol{\alpha}_{i,i} + \sum_{j:\,(i,j)\in E} \boldsymbol{\alpha}_{j,j} \Big) \tag{9}$$

**Neigh-Avg(E).**

$$h_i^{(1)} = \frac{1}{1 + \deg_{\text{in}}(i)} \Big( h_i^{(0)} + \sum_{j:\,(i,j)\in E} h_j^{(0)} \Big) =$$
$$= \frac{1}{1 + \deg_{\text{in}}(i)} \Big( x_{V,i} + \sum_{j:\,(i,j)\in E} x_{V,j} \Big) =$$
$$= \frac{1}{1 + \deg_{\text{in}}(i)} \Big( \boldsymbol{\alpha}_{i,i} + \sum_{j:\,(i,j)\in E} \boldsymbol{\alpha}_{i,j} \Big) \tag{10}$$

Outputs $h_i^{(1)}$ are then fed in input to a logistic regression model, regularised as illustrated in Appendix C.4.1. Before that, they are averaged-pooled in the case of response-wise predictions tasks.

### D.1.2  EXPERIMENTAL CHARM FORM

Throughout all our experiments, CHARM implements the following msg-passing equation:

$$h_i^{(t+1)} = \text{up}_t\Big(\Big[h_i^{(t)} \mid \frac{1}{\deg_{\text{in}}(i)} \sum_{j:\,(i,j)\in E} \text{msg}_t\big(\big[h_j^{(t)} \mid x_{E,(i,j)}^{\tau} \mid p_{i,j}\big]\big)\Big]\Big). \tag{11}$$

Initial node features always include "reflexive attention", i.e., that a token pays to itself. When additionally including activations (CHARM (att+act-*)) we employed an additionally encoder to preprocess these. The output of this module is concatenated to the original attention features before message passing takes place. Note that, in all our experiments, for computational reasons and in alignment with the computational flow of Lookback Lens, we remove all connections from prompt to prompt tokens.

### D.1.3  OPTIMIZER AND SCHEDULERS

For all datasets and tasks, we use the AdamW optimizer Loshchilov & Hutter (2019). We experimented with two learning rate schedulers (see Table 6): "Reduce On Plateau" and "Cosine Annealing with Warmup", where warmup spanned 10% of the total training steps. The scheduler yielding the best validation performance was selected.

### D.2  CODE IMPLEMENTATION

Thee implementation of CHARM was realised by means of PyTorch (Paszke et al., 2019) and PyTorch Geometric (Fey & Lenssen, 2019) (available respectively under the BSD and MIT license). We performed hyperparameter tuning using the Weight and Biases framework (Biewald, 2020). For baselines and models running logistic regression, we resorted to the implementation exposed by the Sci-Kit Learn library (BSD license). LapEig required also running the PCA dimensionality reduction technique; we invoked the python implementation from the same library.

### D.3  EXPERIMENTAL RESOURCES AND ARTEFACTS

We ran our all our experiments on NVIDIA L40 GPUs. The two employed LLMs were both accessed via Hugging Face python API, in particular:

- LLaMa-2-7b-chat (Touvron et al., 2023) (License: LLaMa 2 Community License). Accessed at https://huggingface.co/meta-llama/Llama-2-7b-chat-hf.
- Mistral-7b-instruct (Jiang et al., 2023) (License: Apache-2.0). Accessed at https://huggingface.co/mistralai/Mistral-7B-Instruct-v0.3.

## E  VISUALISATIONS

We provide here two sample visualisations of the constructed computational graphs in input to our CHARM architecture. These illustrate two *test* data points in the NQ dataset and are reported in Figures 5 and 6, where we show the full computational graph (left) along with a zoom-in on the response tokens (right).

These visualisations arrange prompt tokens on the left, and response tokens on the right. Edges are drawn in a way that their thickness and transparency is proportional to the corresponding attention scores, averaged, for illustrative purposes, across layers and heads. Response tokens have their border coloured according to the ground truth — red: the token is in an hallucinated passage; blue: otherwise. Their interior is filled, instead, with a colour that conveys the model prediction — "more red": the token is more likely to be in an hallucinated passage; "more blue": more likely to be in a correct, non-hallucinated passage. Output scores for our model are matched to this colour map linearly after a min-max normalisation. Note the absence of prompt-to-prompt edges, as they are neglected in our experiments as explained at the end of Appendix D.1.2.

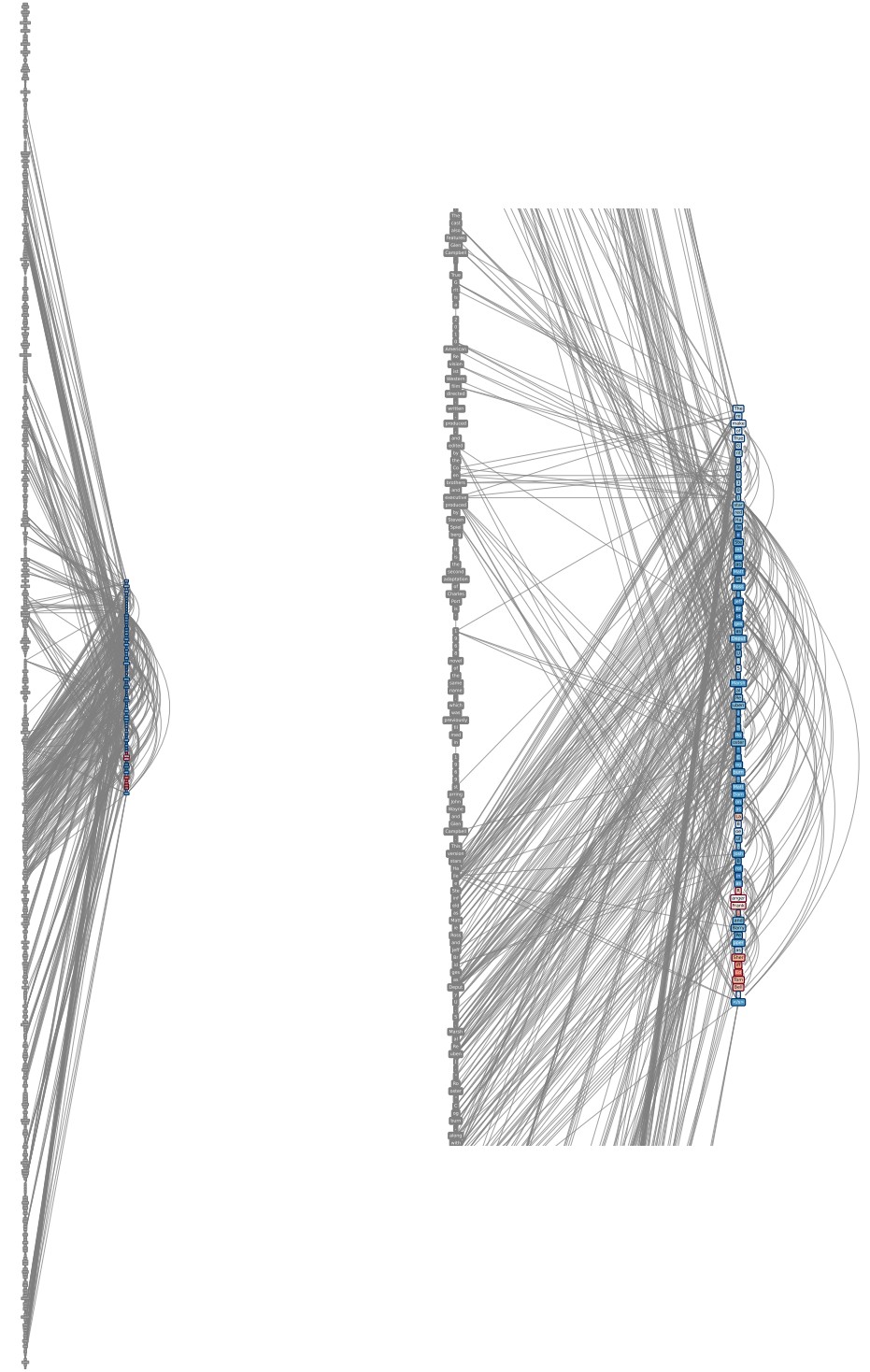

Figure 5: Visualisation of test sample 1590, along with token-wise labels and predictions.

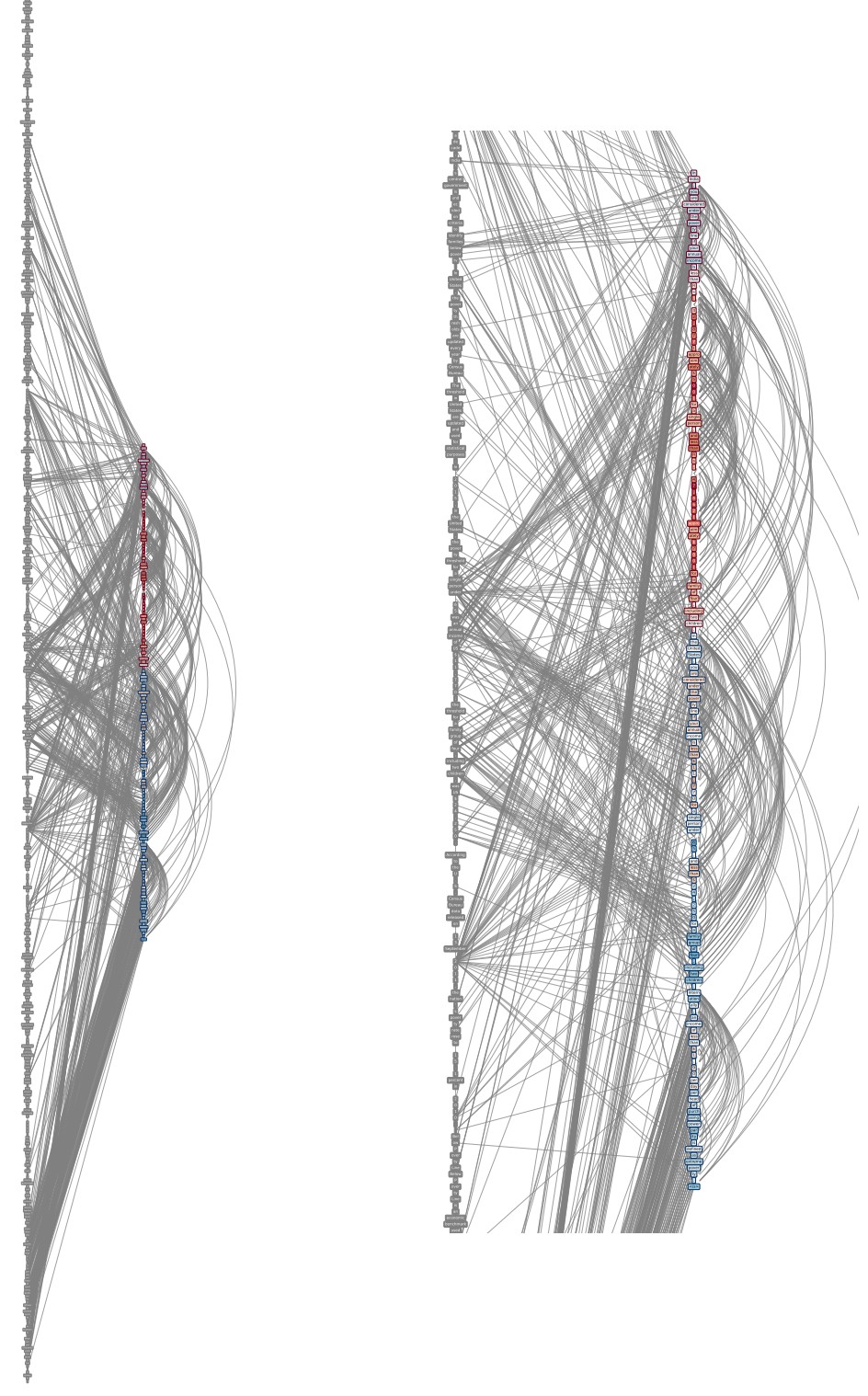

Figure 6: Visualisation of test sample 2354, along with token-wise labels and predictions.

## F  LARGE LANGUAGE MODEL (LLM) USAGE

We employed large language models (LLMs) to support the writing process, specifically for improving clarity in technical explanations, refining grammar and style, and enhancing overall readability. LLMs were also used to a limited extent to aid the process of finding related works. All research contributions, including the design of experiments, data analysis, and conclusions, are entirely our own. The LLMs were used strictly as writing aids to improve presentation quality, not for generating research content or shaping the substance of our work.

