# OpenReview forum: "Neural Message-Passing on Attention Graphs for Hallucination Detection"
_ICLR.cc/2026/Conference — ICLR 2026 Poster_

### Official Review · Reviewer_ffYC · 2025-10-24

**Soundness:** 3
**Presentation:** 4
**Contribution:** 3
**Rating:** 8
**Confidence:** 4

**Summary:**

This paper introduces CHARM (Catching Hallucinated Responses via Message-passing), a unified framework for hallucination detection in LLMs. CHARM represents attention maps and residual activations jointly as attributed graphs, where tokens are nodes and attentional flows define edges. Each node and edge is enriched with features derived from activations and attention scores. Hallucination detection is then formulated as a graph learning problem, addressed via a Graph Neural Network (GNN). The paper shows that CHARM subsumes existing heuristics such as Lookback Lens and LLM-Check and empirical superiority across multiple benchmarks. Experiments cover token-level and response-level hallucination detection across diverse datasets, with comprehensive ablations on graph structure, sparsity thresholds, and zero-shot transfer. CHARM consistently outperforms baselines and shows that incorporating structured, graph-based computation leads to measurable gains over attention- or activation-based heuristics.

**Strengths:**

- The authors presents a novel and unified framework that models LLM computational traces as attributed graphs, effectively integrating both attention scores and activation signals into a single structured representation.

- The proposed approach contributes a new methodological direction by connecting graph neural networks and hallucination detection, opening possibilities for broader applications in explainable AI and reliability of LLMs.

- It provides strong theoretical grounding by formally proving that CHARM subsumes well-known heuristic detectors such as Lookback Lens and LLM-Check, demonstrating clear expressive generality.

- The experimental evaluation is comprehensive, covering both token-level and response-level hallucination detection tasks across five datasets with diverse characteristics.

- CHARM consistently outperforms all baseline methods, including heuristic and activation-based detectors, showing clear empirical advantages in predictive performance.

- The ablation studies and analyses are thorough, investigating the role of graph structure, sparsity thresholds, message passing, and zero-shot transfer in a systematic and transparent manner.

- The paper is well written and organized, with clear mathematical formulations, logical flow, and illustrative figures that make the method easy to understand.

- The results are reported with honesty and nuance, including cases where CHARM’s performance varies in zero-shot transfer scenario, which enhances the paper’s credibility.

**Weaknesses:**

- The performance drop observed when combining activations with attention maps (Table 1) indicates that CHARM may be sensitive to feature selection and layer choice. A deeper discussion on this sensitivity, along with possible mitigation strategies, would strengthen the paper.

- The current framework still relies on heuristic choices to determine which layer of activations to include in the attributed graphs. Developing a more systematic or data-driven approach to layer selection would make the framework more complete and principled.

- There are a few minor writing issues; for instance, inline citations in the first paragraph of the Related Work section should be placed in parentheses.

**Questions:**

- How consistent are the results across different activation layers beyond layers 24 and 32? Could the model be extended to dynamically identify and select the most informative layers during training?

- What mitigation strategies could prevent the degradation in hallucination detection accuracy observed when incorporating additional features, such as activations, into the attributed graph?

---

> ### Author Response · Authors · 2025-11-21
> **Response to reviewer ffYC**
>
> We thank the reviewer for their positive feedback. We are glad they found our method novel and appreciated both the theoretical analysis and the comprehensive experimental evaluation, including the ablation studies. Below, we address their comments and questions.
>
> ---
>
> ***"The current framework still relies on heuristic choices to determine which layer of activations to include in the attributed graphs. Developing a more systematic or data-driven approach to layer selection would make the framework more complete and principled. [...] How consistent are the results across different activation layers beyond layers 24 and 32? Could the model be extended to dynamically identify and select the most informative layers during training?"***
>
> We thank the reviewer for raising this point. As with many learning-based detection methods (e.g., probing approaches), it is standard practice to select layers by optimising performance on a validation set. Nonetheless, we agree that developing a more systematic or data-driven procedure for layer selection could further strengthen the framework. We view this as a valuable direction for future work.
>
> To further test the layer-level consistency, we conducted additional experiments using the same activation layers as in our probing baselines—specifically layers 24, 28, and 32. In particular, we evaluated performance on the Math dataset using activations from layer 28, complementing the previously reported results for layers 24 and 32 in the paper. Additionally, we evaluated performance using all three layers together, to see if the model can dynamically identify and select the most informative layers during training.
>
> The complete results are presented below:
>
>  | Method              | Math Test AUROC | Math Test AUPR |
>  |---------------------|------------|-----------|
> | CHARM (att+act-24)  | 80.8±0.7 | 83.1±0.7 |
> | CHARM (att+act-28)  | 81.2±1.0 | 83.4±1.3 |
>  | CHARM (att+act-32)  | 81.7±0.2 | 83.8±0.3 |
>  | CHARM (att+act-24+28+32)  | **82.7±0.1** | **84.0±0.5** |
>
> These results indicate that CHARM’s performance is relatively stable across layers 24, 28, and 32, with only modest variation. This suggests that the method is not overly sensitive to the specific choice of activation layer within this range. Additionally, we observe that using all three layers together may improve CHARM’s performance, increasing AUROC by 1.0% compared to the best single-layer configuration, hinting towards the reviewer's suggestion that the model could be extended to dynamically identify and select the most informative layers during training. We thank the reviewer for this suggestion, if they deem it relevant, we would be happy to include these preliminary results in the revised manuscript.
>
> ---
>
> ***"There are a few minor writing issues; for instance, inline citations in the first paragraph of the Related Work section should be placed in parentheses."***
>
> We thank the reviewer for the careful attention to detail. We have corrected this minor issue in the revised manuscript.
>
> ---
>
> ***"The performance drop observed when combining activations with attention maps (Table 1) indicates that CHARM may be sensitive to feature selection [...] What mitigation strategies could prevent the degradation in hallucination detection accuracy observed when incorporating additional features, such as activations, into the attributed graph?"***
>
> This is an insightful question, and we thank the reviewer for raising it. In relation to the observed performance drop, we suspect that these degradations may attenuate when more annotated data are available for training, while, either way, standard model selection can also include whether to use certain features or not, and detect good model configurations. We believe that the design of adaptive selection strategies also represents a natural and promising direction for future work. As an initial step, we suggest exploring gating mechanisms that dynamically weight or select the most informative features, which may help mitigate degradation in accuracy when more than one is introduced.

---

> > ### Comment · Reviewer_ffYC · 2025-11-24
> >
> > I thank the authors for their thorough and thoughtful response. Including experimental results with different layer selections, as well as a discussion of potential directions to mitigate performance degradation when incorporating additional features, would further strengthen the final manuscript. As I have already assigned a high score to this paper, I will keep my score unchanged.

---

### Official Review · Reviewer_A2Ck · 2025-10-29

**Soundness:** 3
**Presentation:** 3
**Contribution:** 4
**Rating:** 8
**Confidence:** 5

**Summary:**

The paper describes using a message passing network to use the attention of a transformer to hallucination detection. The method description is straightforward but the implementation and needed input and engineering work is complicated. Experiments across diverse benchmarks (contextual, factual, reasoning) and LLMs (LLaMa-2, Mistral) show that CHARM consistently outperforms existing methods. They evaluation on Natural Qeustions and reasoning datasets.

**Strengths:**

the paper uses internal signal with a well motivated method for attribution. I find this elegat and novel contribution to the best of my knowledge.

The method via GANs moves from fixed, hand-crafted heuristics  to a learnable, expressive model. This allows CHARM to capture more complex and none obvious patterns in the computation that may signal a hallucination.

very good evaluation, very much appriciated, looks good.

The finding that optimal features (attention-only vs. attention+activations) depend on the type of hallucination is a valuable insight.

open sourcing the code very much appriciated.

**Weaknesses:**

Like all methods that rely on internals of methods so CHARM too requires "white-box" access to the model's internal activations and attention matrices.

While GNN inference is reported as fast, it has quite some overhead and implmentation.

**Questions:**

How does the performance (both accuracy and latency) of CHARM degrade as sequence length increases into the tens of thousands of tokens?

The paper uses activations from a single, pre-selected layer (was it layer 24?). Why choosing a single layer? Could performance be improved by integrating activations from multiple layers ? did you try?

The conclusion mentions integrating logits as a future direction. How do the you envision this?

do the you believe the "signature" of a hallucination in the attention graph is universal?

---

> ### Author Response · Authors · 2025-11-21
> **Response to reviewer A2Ck**
>
> We thank the reviewer for their positive feedback. We are pleased that they found our method well-motivated, elegant, and a novel contribution. We also appreciate their recognition of the quality of our evaluation. Below, we address their comments and questions.
>
> ---
>
> ***"Like all methods that rely on internals of methods so CHARM too requires "white-box" access to the model's internal activations and attention matrices."***
>
> We thank the reviewer for raising this point. We agree that CHARM, like all approaches that operate on model internals, requires “white-box’’ access to the model’s activations and attention patterns. We note, however, that as acknowledged by the reviewer, many strong existing baselines make the same assumption: probing methods rely on internal activations, while methods such as LLM-Chk, LAPEig, and Lookback Lens require access to attention matrices. Our method operates under the same assumptions yet achieves consistently better performance in our experiments.
>
>  ---
>
> ***"While GNN inference is reported as fast, it has quite some overhead and implmentation [sic.]."***
>
> We thank the reviewer for raising this point. We note that, while our formulation can introduce some implementation overhead, but also that treating attention matrices as graphs directly unlocks the direct use of standard, well maintained codebases for learning on graphs such as Pytorch Geometric, e.g. supporting off-the-shelf GNN architectures without additional effort.
>
> ---
>
> **"How does the performance (both accuracy and latency) of CHARM degrade as sequence length increases into the tens of thousands of tokens?"***
>
> To address the reviewer’s question regarding CHARM’s performance and latency on long sequences, we present again our results (including run-time) on the CNN dataset, this time as a function of sequence length. These results are now added to our revised version, to Appendix C3 as Figures 3 and 4.
>
> We find that CHARM’s performance does not degrade with increasing sequence length; in fact, the highest AUROC is achieved in the longest-sequence bin. Overall, these results suggest that CHARM is not particularly sensitive to sequence length and that it finds effective applicability also on longer sequences.
>
> ---
>
> ***"The paper uses activations from a single, pre-selected layer (was it layer 24?). Why choosing a single layer? Could performance be improved by integrating activations from multiple layers ? did you try?"***
>
> We thank the reviewer for raising this interesting point. We note that it is common practice in learning-based detection methods to focus on a *single* layer selected via validation.
>
> Following the reviewer’s suggestion, we ran additional experiments on the MATH dataset and report (below) results using activations from layers 24, 28, and 32, both individually and combined together, namely, concatenated as node features.
>
>  | Method              | MATH AUROC | MATH AUPR |
> |---------------------|------------|-----------|
>  | CHARM (att+act-24)  | 80.8±0.7 | 83.1±0.7 |
>  | CHARM (att+act-28)  | 81.2±1.0 | 83.4±1.3 |
> | CHARM (att+act-32)  | 81.7±0.2 | 83.8±0.3 |
>  | CHARM (att+act-24+28+32)  | **82.7±0.1** | **84.0±0.5** |
>
> We observe that using multiple activations together generally improves CHARM's performance, increasing AUROC by 1.0% compared to the best single-layer configuration. We thank the reviewer for this insightful suggestion. We included this result in the revised manuscript (Appendix C.2, Table 7)
>
> ---
>
> ***"The conclusion mentions integrating logits as a future direction. How do the you envision this?"***
>
> We note that any token-level signal can be incorporated into CHARM, since our graph-based framework naturally supports adding such information as node features. In this light, integrating logits as node features is a natural extension, and we hypothesise that doing so could enhance CHARM’s performance by including complementary signals.
>
> ---
>
> ***"do the you believe the "signature" of a hallucination in the attention graph is universal?"***
>
> We thank the reviewer for raising this point, and think this is a very interesting question. As an initial step towards answering it, we kindly refer the reviewer to Table 5 in the paper, where we conducted an experiment evaluating cross-dataset generalization of CHARM by training on one dataset and testing on another zero-shot.
>
> We consistently observed meaningful generalisation (AUC > 0.5), and in both cases, CHARM outperformed standard probing methods. We interpret this as a positive sign towards shared structural graph patterns across datasets. We thank the reviewer for raising this point and view deeper exploration of this direction as a natural and promising avenue for future work.

---

### Official Review · Reviewer_Ku87 · 2025-11-01

**Soundness:** 3
**Presentation:** 3
**Contribution:** 3
**Rating:** 6
**Confidence:** 5

**Summary:**

In this paper, the authors analyze hallucination detection using LLM-internals by employing message passing over a unified attributed graph, incorporating its activations and attention scores. In detail, the graph is set up by assuming tokens as nodes, edges representing the direction of tokens attending to future tokens in a causal manner, the node attributes represented by on-diagonal attention scores and activations, and edge attributes represented by off-diagonal attention scores. The paper proposes Charm, by using a Graph Neural Network over this graph, with token-level hallucination detection cast as a node classification task, and response level detection cast as a graph classification task.

**Strengths:**

1) The paper is well-written and introduces the core components of the proposed method in a clear and concise manner. The paper presents a structured and principled framework to detect hallucinations, by using a GNN to learn from an attribution graph that combines attention scores alongside activation states.

2) With the abstracted framework introduced, the authors show that prior techniques such as LookbackLens and LLM-Check can be represented by specific design choices and instantiations of the GNN used over the general attribute graph.

3) Through empirical evaluations on both token-level hallucination detection and response-level detection, Charm is shown to achieve good empirical performance, and achieves significant improvements over existing baseline. Furthermore, the method is shown to require fairly low memory overheads with sparsification, without inheriting a large drop in performance simultaneously.

**Weaknesses:**

1) The proposed method is considerably more complex to deploy and utilize when compared to baselines: it involves building a potentially large graph for each generation with the number vertices being the same as the maximum token length, and then extracting all attention scores and activations for all layers. This is of course followed by supervised training the GNN on a set of truthful and hallucinated samples, whose distribution is close to the test samples seen at inference time. On the other hand, several baseline methods such as SelfCheck, INSIDE, LLMCheck do not utilize any trainable network components. Furthermore, given that CHARM explicitly trains the GNN, other training based methods such as ITI [1] form a strong baseline of comparison, since the degree of discriminability can be adaptively set for detection.


2) The attribution graph is dense prior to sparsification, and quadratic in the sequence length of  N tokens: $L * H * N * (N+1)/2$ attention scores are computed for an LLM of L layers and H attention heads per layer. While Table-4 does indicate that good performance is achieved even at high-sparsity levels, it is unclear how this would scale for longer sequence lengths. Could the authors kindly report the computation run-time and memory overheads over some fixed sequence lengths such as 1K, 10K and 100K? Furthermore, what is the impact on the generalization of the trained GNN when the context size potentially changes by a large manner at test time?


3) Evaluation Metrics: The empirical evaluation relies exclusively on AUROC and AUPR. While these metrics give a good overall view of performance, in practice it is crucial to achieve high TPR at low FPR rates. Particularly with hallucination detection, it is quite important to report standard detection metrics such as TPR at 5% or 10% FPR and F1 score, to help assess and compare the efficacy of the proposed method?


4) Furthermore, the main experimental evaluations are reported only on LLaMa-2-7B and Mistral-7B, which are slightly dated. The evaluations could be significantly improved by including newer LLMs such as Llama3/4, Gemma and QwQ models. This would also help establish the generality of the method proposed.


5) The specific Response level detection datasets used in the paper (Movies, Winobias, Math) are not commonly used for benchmarking hallucination detection. More standard and established datasets such as HaluEval, TruthfulQA, FAVABench or RagTruth could have been utilized instead.


[1] Inference-Time Intervention: Eliciting Truthful Answers from a Language Model, Li et al., NeurIPS 2023

**Questions:**

Kindly refer to the questions mentioned in the weaknesses section above.

---

> ### Author Response · Authors · 2025-11-21
> **Response to reviewer Ku87 (1/2)**
>
> We are very thankful for the thorough review and glad the reviewer appreciated our work in many ways. We respond to their questions below.
>
> ---
>
> ***“The proposed method is considerably more complex to deploy and utilize [...] several baseline methods such as SelfCheck, INSIDE, LLMCheck do not utilize any trainable network components [...] other training based methods such as ITI [1] form a strong baseline of comparison, since the degree of discriminability can be adaptively set for detection.”***
>
> The reviewer notes the fact that CHARM requires undergoing a training phase, contrary to some baselines. We believe this is a fair point, but would like to respectfully remark how the best performing methods, such as Activation Probes or Lookback Lens, all require parameter fitting. A noteworthy case is that of LLM-Check, which we found to generally underperform out-of-the-box. Interestingly, we experimented with training a simple linear prediction head on top of LLM-Check’s features (LLM-Check++) and this was observed to significantly boost the performance of the original method. This underscores a fundamental, yet important, tradeoff between performance and simplicity of model parameterisation, and we will make sure to highlight this aspect clearly in our next paper revision.
>
> Regarding the ITI method, we thank the reviewer for the pointer. We will make sure to cite this method in the next revision, especially given its complementary focus on steering. As for the detection strategy, we found it similar to activation probes, and we have not prioritised implementing it on our datasets as of now. If the reviewer deems it crucial, we would be happy to make further efforts in this direction. We would also be glad if they could better clarify what they mean by “the degree of discriminability can be adaptively set for detection”.
>
> ---
>
> ***[...] good performance is achieved even at high-sparsity levels, it is unclear how this would scale for longer sequence lengths. Could the authors kindly report the computation run-time and memory overheads over some fixed sequence lengths such as 1K, 10K and 100K?***
>
> The reviewer asked about the computational run-time and memory overhead of CHARM with sequence lengths 1k, 10k, 100k. We agree this is an important question to ask, but also note that we are not aware of relevant datasets with 10k or 100k long sequences. Our considered CNN dataset, however, contains samples with lengths ranging from ~ 500 to ~ **2.7k tokens**, and we take this as a reference dataset to experiment with. In particular, we measure the inference run-time of CHARM and the memory consumed by computational trace graphs on GPU as a function of the corresponding sequence length. The obtained trends are depicted in our revised manuscript, in Appendix C.3 (Figure 3). We observe that, even for the longest sequences with more than 2.5k tokens, the median runtime is ~ 1.2 ms. This value is only marginally higher than that for the shortest sequences, with less than 1.1 ms median inference time for sequences with a median of ~ 500 tokens. Similarly, the memory footprint for the largest computational graphs is reasonably contained, with a median of less than 130 MB for sequences of around 2.5k tokens, compared to a median footprint of ~ 80 MB for the shortest ones.
>
> Importantly, not only are these values contained, we also do not observe any quadratic “explosion” in the observed trends. This underscores, once more, the importance of our sparsification approach, which allows us to swiftly deal with (quadratic) attention maps.
>
> ---
>
> ***“what is the impact on the generalization of the trained GNN when the context size potentially changes by a large manner at test time?”***
>
> The reviewer raises an interesting point. To experimentally elicit the hypothesised scenario, we focussed on the CNN dataset, and produced a new non-uniform split whereby test samples are taken as the top 20% longest sequences, the rest considered for training and validation sets. This way, training and model selection is run on sequences only up to ~ 1.2k tokens, while the model is tested on sequences of length ranging from this value to ~ 2.7k tokens. Results are reported in the table below, comparing CHARM with other prominent baselines:
>
> | Model  | CNN Test AUROC (longer seqs)   | CNN Test AUPR (longer seqs)   |
> |---|----|---|
> | Act-24         | 71.64  | 16.25 |
> | Act-28         | 70.33  | 15.98   |
> | Act-32         | 68.43  | 14.10 |
> | Lookback Lens    | 73.94  | 18.85  |
> | CHARM            | **74.50 ± 0.67**    | **21.59 ± 1.18**  |
>
> We observe that CHARM still outperforms methods in comparisons on both metrics, and that its performance on unseen sequence lengths remains stable, only marginally lower than what previously obtained on uniform splits.
>
> We thank the reviewer for raising this interesting point which allowed us to further strengthen our work with these new results. We have included them in our new revision (see Appendix C.3, Table 8).

---

> ### Author Response · Authors · 2025-11-21
> **Response to reviewer Ku87 (2/2)**
>
> ***“[...] report standard detection metrics such as TPR at 5% or 10% FPR and F1 score, to help assess and compare the efficacy of the proposed method”***
>
> The reviewer appreciated the joint use of AUROC and AUPR, but also asked about other metrics such as TPR@5%FPR, TPR@10%FPR and F1.
>
> We ran a new focussed experiment on the NQ dataset, retraining and evaluating CHARM, Activation Probes, and the Lookback baseline with these additional metrics. Results are reported in the table below:
>
> | Model            | Test TPR@5    | Test TPR@10       | F1                   |
> |------------------|----------------------|---------------------------|---------------------------|
> | ACT (24)         |   0.230           |   0.356                |   0.107                |
> | ACT (28)         |   0.217           |   0.329                |   0.136                |
> | ACT (32)         |   0.149           |   0.245                |   0.145                |
> | Lookback Lens    |   0.190           |   0.306                |   0.114                |
> | CHARM            |   **0.254 ± 0.040**           |   **0.401 ± 0.021**                |   **0.360 ± 0.034**                |
>
> We observe that CHARM attains superior performance w.r.t. baselines on all metrics, sometimes considerably. For completeness, we note that the F1 score has been calculated with the standard threshold of 0.5 for all methods.
>
> ---
>
> ***“evaluations could be significantly improved by including newer LLMs such as Llama3/4, Gemma and QwQ models”***
>
> The reviewer is raising an interesting point: it would be insightful to validate CHARM’s effectiveness on LLMs more recently released. While we agree that this would strengthen our claims, we are not aware of ready-to-use publicly available datasets targeting the mentioned LLMs that we could easily integrate now. The already considered datasets do not target these LLMs either, and adapting them would require particular efforts, e.g., in regenerating annotations. This is, however, a valuable endeavour we will consider for future research, and we thank the reviewer for pointing this out.
>
> ---
>
> ***"More standard and established datasets such as HaluEval, TruthfulQA, FAVABench or RagTruth could have been utilized instead"***
>
> The reviewer recommended considering other datasets, and we appreciate the suggestion. During this week, we made an effort to extend our evaluations to the RAGTruth suite and, in particular, its QA dataset, targeting the Mistral-7B-instruct LLM. We employed the officially provided train/test split. From the training set, we uniformly sampled 20% of the data to form a validation set utilised to tune hyperparameters for all considered approaches. This constructed RAGTruth QA dataset is similar to our previously considered NQ, in that it casts a contextual hallucination detection task and is annotated at the level of single tokens. We thus set the task at a token-level granularity.
>
> As above, we compare CHARM to Lookback Lens and Activation Probes. Results are reported below:
>
> | Method                                | RAGTruth QA Test AUROC | RAGTruth QA Test AUPR |
> |----------------------------------------|------------|--------------------|
> | Act-24                                 | 91.7      | 61.6              |
> | Act-28                                 | 90.5      | 57.7              |
> | Act-32                                 | 86.2      | 45.0              |
> | Lookback Lens                          | 91.9      | 63.5              |
> | CHARM     | **93.0 ± 0.1** | **67.6 ± 0.00**   |
>
> On this dataset, CHARM remarkably outperforms baselines with notable gaps especially in terms of AUPR. We thank the reviewer for prompting us to explore these additional benchmarks, which allowed us to strengthen our claims further.

---

### Official Review · Reviewer_9j71 · 2025-11-04

**Soundness:** 3
**Presentation:** 2
**Contribution:** 3
**Rating:** 6
**Confidence:** 3

**Summary:**

This paper introduces CHARM, a framework that recasts LLM hallucination detection as a graph learning problem. It models an LLM's internal computational traces, specifically attention scores and activations, as a single attributed graph where tokens serve as nodes and attention maps define the edges. This formulation elevates the task from relying on isolated heuristics to a principled problem effectively tackled by a Graph Neural Network (GNN).

**Strengths:**

1. The paper proposes a novel approach to hallucination detection by modeling LLM computational traces (attention and activations) as attributed graphs. This formulation provides a unified view of different internal signals and allows for the application of established graph neural network architectures to the problem.
2. The proposed method is supported by both theoretical analysis and empirical evaluation. The authors show that their framework is expressive enough to subsume existing attention-based heuristics. The experiments, conducted across different benchmarks and detection granularities, demonstrate performance gains over several baselines.
3. The paper is well-organized and clearly written.

**Weaknesses:**

1. The choice of LLM layer corresponding to the activation values is not clearly justified. For example, in Tables 1 and 2, some baseline methods report results from multiple layers, while CHARM only uses the layer 24. It would be better to include results from different layers to provide a more comprehensive analysis. Moreover, as mentioned at the end of Section 5.2, CHARM uses the layer 32 only for the Math dataset. Does this imply that performance is highly sensitive to layer selection?
2. The paper adopts a relatively simple propagation mechanism, whereas in previous studies on GNNs, more sophisticated approaches such as GCN or GAT are widely used. The authors should clarify the rationale for not employing these advanced graph propagation methods.
3. The paper lacks visualizations. It would be helpful to include concrete examples illustrating the constructed graph, such as visualizing nodes of varying sizes and edges with different thicknesses, or employing heatmaps and other effective visualization methods. Such visualizations could greatly enhance the interpretability of the proposed method.

**Questions:**

See the Weaknesses.

---

> ### Author Response · Authors · 2025-11-21
> **Response to reviewer 9j71 (1/2)**
>
> We thank the reviewer for their positive feedback. We are glad they recognised the novelty of our graph-based approach, which unifies attention and activation signals and enables the use of established GNN architectures for hallucination detection. We also appreciate their acknowledgement of our theoretical and empirical contributions. Below we address their comments and questions.
>
>  ---
>
> ***"The choice of LLM layer corresponding to the activation values is not clearly justified. For example, in Tables 1 and 2, some baseline methods report results from multiple layers, while CHARM only uses the layer 24. It would be better to include results from different layers to provide a more comprehensive analysis. Moreover, as mentioned at the end of Section 5.2, CHARM uses the layer 32 only for the Math dataset. Does this imply that performance is highly sensitive to layer selection?"***
>
> We thank the reviewer for raising this point. In our work, we chose layer 24 for CHARM when enhanced with activation signals, simply based on the performance of activation-based baselines, which generally performed best on layer 24. Regarding the Math dataset: this was the only one where activation-based baselines performed better on layer 32, and, because of this reason, for completeness we *also* reported the performance of our method endowed with these different features inline in text.
>
> In general, we expect that a specific selection of the activation layer for CHARM would only improve our results. In any case, following the reviewer's suggestion, we have conducted additional experiments on the Math dataset for a better understanding of the sensitivity of CHARM to the layer selection. We obtained results also for layer 28 in addition to the previously reported layers 24 and 32 and report them in full here below:
>
> | Method    | Math Test AUROC | Math Test AUPR |
> |---------------------|------------|-----------|
> | Act-24 |  77.7  |  77.5  |
> | Act-28 |  78.1  |  77.8  |
> | Act-32 |  76.6  |  77.9  |
> | CHARM (att+act-24)  | 80.8±0.7 | 83.1±0.7 |
> | CHARM (att+act-28)  | 81.2±1.0 | 83.4±1.3 |
> | CHARM (att+act-32)  | **81.7±0.2** | **83.8±0.3** |
>
> The performances of our method are observed to be relatively stable and higher than activation probes. Importantly, they demonstrate that the performance of CHARM when enhanced with activations is not overly sensitive to layer choice.
>
> ---
>
> ***"The paper adopts a relatively simple propagation mechanism, whereas in previous studies on GNNs, more sophisticated approaches such as GCN or GAT are widely used. The authors should clarify the rationale for not employing these advanced graph propagation methods."***
>
> We appreciate the reviewer’s suggestion to explore more sophisticated message-passing (MP) mechanisms, and we have conducted additional experiments accordingly. Before presenting these results, we would like to clarify that the primary objective of our work was to demonstrate the overall effectiveness of a graph-based approach. For this reason, we did not focus on selecting or tuning the most advanced message-passing mechanism; instead, we adopted a general, yet expressive MP formulation, as defined in Equation (2) of our paper. Indeed, we adopt the formulation inspired by [1], shown in the same paper to subsume many others.
>
> Either way, following the reviewer’s recommendation, we ran further experiments with two other message-passing architectures: GAT and GATv2. We did not include GCN, although suggested, because it does not naturally incorporate edge features, an essential component of our proposed graph formulation. The results for GAT and GATv2 on the NQ dataset are presented in the table below.
>
>  | Method     | NQ Test AUROC | NQ Test AUPR |
> |---------------------|----------|---------|
> | CHARM (GAT)  | 74.6±1.1 | 38.9±1.6 |
> | CHARM (GATv2)  | **74.9±0.8** | 38.8±0.7 |
> | CHARM (Our MP)  | 74.8±0.6 | **40.3±1.7** |
>
> These results indicate that CHARM’s performance remains broadly consistent across different MP mechanisms. While GATv2 yields a comparable AUROC, our simple MP mechanism attains the best AUPR. Overall, we view the exploration of more sophisticated MP schemes as a natural and promising direction for future work on hallucination detection.
>
> **References**
>
> [1] Gilmer et al., Neural Message Passing for Quantum Chemistry, ICML 2017

---

> > ### Author Response · Authors · 2025-11-21
> > **Response to reviewer 9j71 (2/2)**
> >
> > ***"The paper lacks visualizations. It would be helpful to include concrete examples illustrating the constructed graph, such as visualizing nodes of varying sizes and edges with different thicknesses, or employing heatmaps and other effective visualization methods. Such visualizations could greatly enhance the interpretability of the proposed method."***
> >
> > We appreciate the reviewer’s suggestion to include input visualisations that would enhance the interpretability of our approach. To address this, we have added Appendix E in our revised manuscript, where we provide preliminary visualisations for a couple of NQ constructed graphs in input to our model. These arrange prompt tokens on the left, and response tokens on the right. Edges are drawn in a way that their thickness and transparency is proportional to the corresponding attention scores, averaged, for illustrative purposes, across layers and heads. Response tokens have their border coloured according to the ground truth; their interior is filled, instead, with a colour that conveys the model prediction — “more red”: the token is more likely to be in an hallucinated passage; “more blue”: more likely to be in a correct, non-hallucinated passage. Given the number of tokens, visualising the full graphs within the page limits is relatively tough, and we hence provided also zoomed-in depictions around response tokens. In any case, the images are in vectorial format, and it should be possible to manually zoom-in as much as desired.

---

### Author Response · Authors · 2025-11-21
**General response**

We are extremely grateful to all reviewers for their extensive and insightful feedback. We are very glad our work has been positively received by all of them, who found our approach “*novel*” (**9j71**, **A2Ck**, **ffYC**), *theoretically* “*supported*” (**9j71**) and *grounded* (**ffYC**), “*well motivated*” and “*elegant*” (**A2Ck**), “*structured*” and “*principled*” (**Ku87**). The reviewers have appreciated how this “new methodological direction” (**ff6YC**) is *expressive* enough to subsume previous techniques (**9j71**, **Ku87**, **A2Ck**, **ff6YC**). The supporting empirical analysis (**9j71**) has also been found “*comprehensive*” (**ff6YC**) and “*very good*” (**A2Ck**), and to *show clear performance gains* with respect to current methods (**9j71**, **Ku87**, **ffYC**). Reviewers **A2Ck** and **ffYC** have, in particular, positively valued the insights provided by our ablation studies and analyses, deemed “*thorough*”, “*systematic*”, “*transparent*” (**ffYC**) and presented with “*honesty and nuance*” (**ffYC**). Finally, we are glad the paper has been found “*well-organised*” (**9j71**, **ffYC**) and *clearly written* (**9j71**, **Ku87**, **ffYC**), with “*clear mathematical formulation, logical flow and illustrative figures*” (**ffYC**).

While we answer points raised by each single reviewer below in specific responses, here we believe it might be helpful to comment on shared concerns.

---

### **On the choice of the Activation Layer used as an input to CHARM**

Reviewers **9j71**, **A2Ck** and **ffYC** asked about the rationale behind the specific choice of the activation layer utilised in CHARM, whether our approach may be sensitive to this choice, and whether multiple layers can be jointly considered in our architecture.

Beyond referring to specific responses below, here we remark how the choice of the layer for our method has been driven by a simple rationale: we picked layer 24 as this performed, overall, most robustly in activation probing baselines. Generally speaking, the activation layer can be treated as a standard hyperparameter to optimise on a validation set. However, in an effort to provide additional insights to reviewers, we ran a focused experiment on the Math dataset, training (and tuning) our method with all the layer choices considered for probes, that is, 24, 28, 32, as well as all of them together jointly.

CHARM performed robustly across these layer choices, outperforming probes in all cases. Considering all activation layers together, even improved performance.

---

### **On scaling to longer sequence lengths**

Reviewers **Ku87** and **A2Ck** both asked how **CHARM** scales, e.g., in terms of latency, memory consumption, performance, to long sequences.

We note that the considered CNN dataset has samples with length spanning from hundreds to thousands of tokens, up to ~2.7k. We considered this dataset and ran specific experiments measuring the aforementioned quantities as a function of the sequence length.

Results confirmed no performance drops over sequences with thousands of tokens, as well as extremely contained run-times and memory consumptions. These two were observed to scale very favourably in the considered range, underscoring the crucial computational advantage of running neural message-passing on *sparsified* attention graphs.

We additionally ran an experiment where we trained on shorter sequences and tested on longer ones, still on the CNN dataset. Our model scored strong results also in this setup, outperforming baselines.

---

### **Manuscript revision**

Along with the provided rebuttal responses, we are concurrently uploading a revision to our manuscript, which includes new salient experimental results along with additional visualisations. Edits are highlighted in cyan. We hope this, along with our responses, helps clarify reviewers' concerns. We remain open to further questions.

---

### Author Response · Authors · 2025-12-01
**Summary message for re-assigned AC (1/2)**

In an effort to assist the newly assigned Area Chair (AC), below we summarise the outcome of reviews and the following rebuttal discussions.

Overall we are very grateful to the Senior AC, the initially assigned AC and the reviewers for their contributions so far. In the rebuttal process, we answered all their questions and ran additional experiments, this contributing to further strengthening our paper.

---

**Initial assessment**

All the four reviewers provided a positive evaluation of our work, which received initial scores:
- 6 (9j71)
- 6 (Ku87)
- 8 (A2Ck)
- 8 (ffYC)

The reviewers jointly agreed on several positive aspects of our paper and shared interesting comments and questions. We answered them all in our rebuttal, supporting our responses with additional experiments wherever required. Below we report highlighted positive aspects, questions and comments, along with a concise summary of how we addressed these.

---

**Strengths**

Reviewers positively highlighted:
- **Our methodological approach** (*all reviewers*), which was found
  - “novel” (*9j71, A2Ck, ffYC*)
  - theoretically “supported” (*9j71*) and “grounded” (*ffYC*)
  - “well motivated” and “elegant” (*A2Ck*)
  - “structured” and “principled” (*Ku87*)
- **The proved expressiveness of our method** (*all reviewers*), in particular how it allows to subsume existing techniques
- **The empirical analysis**, (*9j71, ffYC, A2Ck*) found:
  - supporting (*9j71*)
  - “comprehensive” (*ffYC*)
  - “very good” (*A2Ck*)
- **The clear performance gains by our method** (*9j71, Ku87, ffYC*)
- **Our ablations and analyses** (*A2Ck, ffYC*), deemed:
  - thorough, systematic, transparent (*ffYC*)
  - to generate valuable insights (*A2Ck*)
- **Our presentation quality** (*9j71, Ku87, ffYC*); the paper was found:
  - “well organised” (*9j71, ffYC*)
  - “clearly written” (*9j71, Ku87, ffYC*)
  - with “clear mathematical formulation, logical flow and illustrative figures” (*ffYC*)
  - with results presented with “honesty and nuance” (*ffYC*)

---

> ### Author Response · Authors · 2025-12-01
> **Summary message for re-assigned AC (2/2)**
>
> **Comments, questions and how we addressed them**
> - **Choosing the activation layers in input to CHARM** (our method): Reviewers *9j71, A2Ck* and *ffYC* asked about the rationale behind this specific choice, whether our approach may be sensitive to it, and whether multiple layers can be jointly considered.
>   - We remarked that the activation layer can be treated as a standard hyperparameter to tune on a validation set.
>   - We ran an experiment on the Math dataset, training (and tuning) our method with all the layer choices considered for probe baselines: 24, 28, 32, as well as all of them together jointly (as asked specifically by reviewer *A2Ck*).
>   - Our method *performed robustly across these layer choices, outperforming probes in all cases, and considering all activation layers together, even improved performance*.
> - **Scaling to longer sequences**: Reviewers *Ku87* and *A2Ck* both asked how CHARM scales, e.g., in terms of latency, memory consumption, performance, to long sequences.
>   - We ran experiments on the CNN dataset (up to 2.7k long samples), measuring the aforementioned quantities as a function of the sequence length.
>   - *No performance drops over sequences with thousands of tokens were observed, as well as extremely contained run-times and memory consumptions*.
> - **Performance on sequences of length unseen at training time**:  Reviewer *Ku87* asked if the model generalises well to longer sequences than those encountered at train time.
>   - On CNN, we additionally ran an experiment where we trained on shorter sequences and tested on longer ones.
>   - Our model scored *strong results also in this setup, outperforming baselines*.
> - **Employing other message-passing primitives**: Reviewer *9j71* wondered about the impact of other Graph Neural Network architectures.
>   - We equipped our method with GAT and GATv2 layers and benchmarked it on the NQ dataset.
>   - *Results showed that CHARM’s performance remains consistent*, indicating its general robustness to the choice of the message-passing layer.
> - **Evaluating CHARM with other performance metrics (TPR at 5% or 10% FPR, F1)** was deemed relevant by reviewer *Ku87*.
>   - We (retrained and) evaluated our method along with prominent baselines on the NQ dataset with the above metrics.
>   - *Our method outperformed baselines also according to the requested metrics*.
> - **Extending evaluations to more recently released LLMs and datasets**: Reviewer *9j71* shared his view that this would have strengthened the manuscript.
>   - We noted that known publicly available datasets do not target newer LLMs, while we made an effort to extend our benchmarking to the RAGTruth benchmark, as recommended by the reviewer. We trained, tuned and evaluated CHARM as well as prominent baselines on its QA dataset.
>   - *Our method significantly outperformed baselines on this dataset*.
> - **Adding graph visualisations**; this was recommended by Reviewer *9j71*.
>   - We produced sample visualisations which show the graphs constructed from the prompt-response attention generated attention maps as well as model token-wise predictions compared to ground-truth annotations.
>   - These visualisations are in the paper revision, Appendix E.
> - **On the additional complexity of a trainable method**: Reviewer *Ku87* highlighted this point in comparison to the simplicity of non-trainable baselines.
>   - We argued how performance improvements are significant, and how even the best baselines fit some parameters on a training set.
>   - To further elucidate this point, we highlighted the noteworthy improvement we obtained by enhancing a non-trainable baseline (LLM-Check) with a final, trainable prediction head.
> - **Other general questions and ideas for improvements** were shared by Reviewers *A2Ck* and *ffYC*, around, e.g., selecting input signals, and on the generality of the extracted hallucination patterns.
>   - We shared our views with these reviewers in their respective rebuttal responses.
>
> Our new paper revision includes these new salient experimental results as well as the new provided graph visualisations.
>
> ---
>
> **Post rebuttal phase**
>
> We believe our responses provided strong evidence to positively address the reviewer’s concerns. Reviewer *ffYC* recognised our response as thoughtful and thorough, and recommended including it in the revised manuscript. They maintained their clearly positive score (8).
>
> We were still awaiting responses from other reviewers when we got informed about the occurrence of data leakage and the interruption of the discussion period.
>
> ---
>
> We hope this summary message will be helpful in making an informed decision. Thank you very much in advance.
>
> \- The Authors

---

### Meta-Review · Area_Chair_cA4B · 2025-12-26

**Summary:**

This paper introduces CHARM, a framework that recasts LLM hallucination detection as a graph learning problem. The approach subsumes prior heuristics and shows consistent empirical gains.
Based on the feedback from reviewers, the decision was made to recommend it for acceptance. We congratulate the authors on their acceptance!
On the other hand, authors should revise the paper taking into account the reviewers' comments, such as the issues and concerns mentioned in Weaknesses.

**Reviewer Concerns:**

Reviewers note sensitivity to layer selection, model complexity, and limited metrics or model coverage. These concerns are largely analytical or engineering-related, acknowledged by the authors, and do not undermine the core methodological contribution.

**Reviewer Scores:**

Reviewers rate soundness and contribution as good to excellent, with two strong accepts and two marginal accepts. There is consensus on novelty, technical correctness, and empirical strength. Overall sentiment supports acceptance, with remaining concerns suitable for future work.

---

### Decision · Program_Chairs · 2026-01-26

Accept (Poster)